# Evidence for a trap-and-flip mechanism in a proton-dependent lipid transporter

Elisabeth Lambert [1,6], Ahmad Reza Mehdipour[2,6], Alexander Schmidt [3], Gerhard Hummer [4,5] & Camilo Perez [1✉]

Transport of lipids across membranes is fundamental for diverse biological pathways in cells. Multiple ion-coupled transporters take part in lipid translocation, but their mechanisms remain largely unknown. Major facilitator superfamily (MFS) lipid transporters play central roles in cell wall synthesis, brain development and function, lipids recycling, and cell signaling. Recent structures of MFS lipid transporters revealed overlapping architectural features pointing towards a common mechanism. Here we used cysteine disulfide trapping, molecular dynamics simulations, mutagenesis analysis, and transport assays in vitro and in vivo, to investigate the mechanism of LtaA, a proton-dependent MFS lipid transporter essential for lipoteichoic acid synthesis in the pathogen *Staphylococcus aureus*. We reveal that LtaA displays asymmetric lateral openings with distinct functional relevance and that cycling through outward- and inward-facing conformations is essential for transport activity. We demonstrate that while the entire amphipathic central cavity of LtaA contributes to lipid binding, its hydrophilic pocket dictates substrate specificity. We propose that LtaA catalyzes lipid translocation by a 'trap-and-flip' mechanism that might be shared among MFS lipid transporters.

[1] Biozentrum, University of Basel, Basel, Switzerland. [2] Center for Molecular Modeling, Ghent University, Zwijnaarde, Belgium. [3] Proteomics Core Facility, Biozentrum, University of Basel, Basel, Switzerland. [4] Institute of Biophysics, Goethe University Frankfurt, Frankfurt am Main, Germany. [5] Department of Theoretical Biophysics, Max Planck Institute of Biophysics, Frankfurt am Main, Germany. [6] These authors contributed equally: Elisabeth Lambert, Ahmad Reza Mehdipour. ✉email: camilo.perez@unibas.ch

Major facilitator superfamily (MFS) transporters are found in all kingdoms of life and move a large variety of molecules across biological membranes[1–8]. Structural characterization of MFS transporters that participate in the uptake of water-soluble molecules and extrusion of drugs has contributed to a broad understanding of their transport mechanism[4,8–17]. However, multiple reports have attributed alternative functions to MFS transporters, such as the translocation of lipids associated with fundamental biological pathways. Some examples include the bacterial lysophospholipid transporter LplT, involved in lipids recycling in Gram-negative bacteria[7,18]; the human transporter MFSD2A, expressed at the blood–brain and blood–retinal barrier, contributing to major uptake of docosahexaenoic acid (DHA)[5,6,19–22]; the human transporters Spns2[23,24], and MFSD2B[25], which contribute to transport of sphingosine 1-phosphate (S1P) in endothelial cells and erythrocytes; and the gentiobiosyl-diacylglycerol transporter LtaA, involved in cell wall synthesis in *Staphylococcus aureus*[26,27]. However, despite their well-described cellular roles, the mechanisms of MFS lipid transporters remain insufficiently understood.

We have previously shown that LtaA is a proton-dependent MFS lipid antiporter[27]. It contributes to the adaptation of *S. aureus* to acidic conditions, common in the skin and nasopharynx of the human host[27–29]. LtaA takes part in the assembly of lipoteichoic acid, a phosphate-rich polymer important for control of bacterial cell division, protection from environmental stress, host cell adhesion, antibiotic resistance, biofilm formation, and immune evasion[30–33]. *S. aureus* lipoteichoic acid displays a polymer of 1,3-glycerol-phosphate repeat units attached to C-6 of the non-reducing glucosyl of the glycolipid gentiobiosyl-diacylglycerol[32–34]. This glycolipid is synthesized at the cytoplasmic leaflet of the membrane by the glycosyltransferase YpfP and is translocated to the outer leaflet by the activity of LtaA[26,27]. The essential role of LtaA in adjusting the pool of glycolipids available at the extracellular side of the membrane makes this protein a central player for lipoteichoic acid assembly and a potential target for drugs aiming to counteract antimicrobial-resistant *S. aureus* strains e.g., methicillin-resistance *S. aureus* (MRSA) and vancomycin-resistant *S. aureus* (VRSA)[31].

Two different general models of transporter-catalyzed lipid translocation have been proposed in the past[35–43]. A "trap-and-flip" model, in which the lipid substrate is retrieved from one leaflet of the membrane, enclosed into a central cavity, and then delivered to the other leaflet[41,44,45], and a "credit-card" model that departs from the classical alternating-access model and involves translocation of the lipid headgroup across a hydrophilic cleft or cavity in the transport protein, while aliphatic chains remain embedded in the membrane[37–39,42,43,46,47]. However, it is not known which of these two models describe better the mechanism of MFS lipid transporters. Answering this question is not only important to understand the basis of the processes catalyzed by these proteins but could also provide a foundation for the design of drugs and/or lipid-linked-bioactive molecules targeting cells or organs expressing pharmacologically relevant proteins from this superfamily.

Until now, a high-resolution structure of outward-facing LtaA, and inward- and outward-facing structures of MFSD2A have been elucidated[21,22,27]. Both transporters display the canonical MFS fold of 12 transmembranes (TM) helices and an amphipathic central cavity that has not been observed in any MFS transporter of water-soluble molecules. The similar architectural features observed in the structures of LtaA and MFSD2A indicate common elements in their transport mechanisms and likely among all MFS lipid transporters. Here, we used cysteine disulfide trapping of outward- and inward-facing LtaA, in combination

with molecular dynamics simulations, mutagenesis analysis, and transport assays in vitro and in vivo, and showed that cycling through outward- and inward-facing conformations is essential for LtaA activity. We demonstrate that LtaA displays membrane exposed lateral openings with distinct functional relevance and characterized the architecture and biochemical properties of the amphipathic central cavity during alternating access. Our results indicate that while the hydrophilic pocket of the amphipathic central cavity dictates substrate specificity, the hydrophobic pocket is only relevant for aliphatic chains' binding. We describe critical mechanistic elements revealing that LtaA adopts a "trap-and-flip" mechanism that might be shared among MFS lipid transporters.

## Results

**Models of inward-facing LtaA and validation by cysteine crosslinking.** To investigate whether LtaA uses a "trap-and-flip" or a "credit-card" mechanism, we first aimed to establish a system that allowed us to perform cysteine disulfide trapping of end-point conformations of LtaA during its transport cycle. The architecture of the previously solved structure of LtaA[27], facilitates cysteine disulfide trapping of outward-facing states, whereas there is no structural information to guide trapping of inward-facing states. Thus, we first generated an inward-facing model of LtaA using "repeat-swap" modeling[48]. Like other transporters from the MFS superfamily, the topology of LtaA comprises two domains, a N-terminal domain (TM1-TM6; domain-1), and a C-terminal domain (TM7-TM12; domain-2), each of which contains two structural repeats with inverted-topology related by a pseudo-rotational twofold symmetry axis parallel to the plane of the membrane (Fig. 1a, b). After swapping the conformations of the inverted repeats observed in the outward-facing structure of LtaA (PDB ID 6S7V)[27,48], we constructed a large set of models in silico that were refined aiming to improve side chains packing, stereochemistry, and modeling scores. The models with the best scores converged to one conformation (Fig. 1c and Supplementary Table 1), which displayed multiple interactions between the extracellular parts of TM1-TM7, TM2-TM11, and TM5-TM8, sealing the entrance to the central cavity (Fig. 1c). In contrast, the cytoplasmic regions of helices TM2-TM11, TM5-TM8, and TM4-TM10, lining the entrance to the central cavity from the cytoplasm, are away from each other about $16.0 \pm 0.1$, $16.0 \pm 0.1$, and $17.6 \pm 0.2$ Å, respectively (Fig. 1c). The helical loop between TM6 and TM7 that connects the N- and C-terminal domains was modeled based on the conformation observed in the outward-facing structure.

An additional inward-facing LtaA model was generated by AlphaFold (AF) (Fig. 1d)[49]. In agreement with the models generated by the "repeat-swap" method, in the AF model, the entrance to the central cavity is sealed by multiple interactions between the extracellular parts of TM1-TM7, TM2-TM11, and TM5-TM8 (Fig. 1d), whereas the cytoplasmic regions of helices TM2-TM11, TM5-TM8, and TM4-TM10, lining the cytoplasmic cavity, are away from each other about 25.0, 25.9, and 23.2 Å, respectively (Fig. 1d). Comparison of the AF model and the "repeat-swap" models reveals a wider opening of the cytoplasmic cavity in the model generated by AF (Fig. 1c, d).

To validate these inward-facing models, we selected pairs of residues among the extracellular regions for which Cβ–Cβ distances were less or close to 8.0 Å, but which present Cβ–Cβ distances of over 12 Å in the outward-facing structure (Supplementary Fig. 1). We excluded those residues that were predicted to be buried or located in flexible regions. Based on these criteria, we identified the pairs F45-T253, A53-T366, and K166-I250 (Fig. 2a and Supplementary Fig. 1), occupying three different

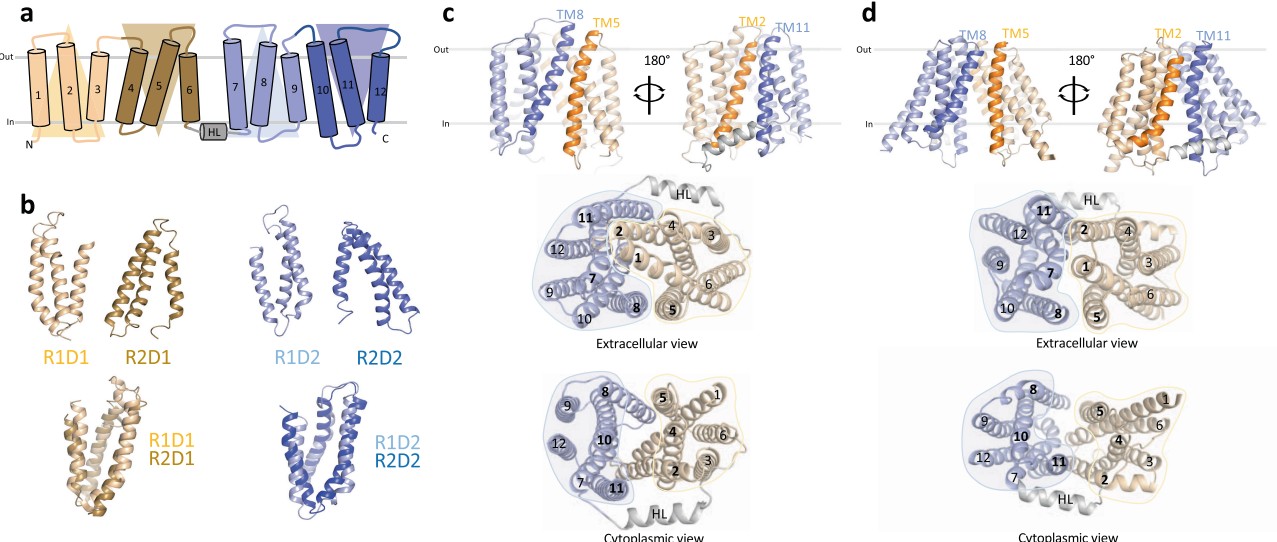

**Fig. 1 Modeling of inward-facing LtaA. a** Topology representation of LtaA. Domain-1 (N-terminal) and domain-2 (C-terminal) are shown in light orange and light blue, respectively. **b** Individual repeat domains as observed in outward-facing LtaA (PDB ID 6S7V), and superposition of inverted repeats (r.m.s.d. = 2.5 Å and 3.0 Å for aligned Cα atoms of R1D1/R2D1 and R1D2/R2D2, respectively). R1D1 and R2D1 indicate the first and second repeats in the N-terminal domain, respectively, whereas R1D2 and R2D2 indicate the first and second repeat in the C-terminal domain, respectively. Colors are according to panel **a**. **c**, **d** Side-views of inward-facing LtaA models generated by "repeat-swap" and by AlphaFold (AF), respectively. The models show TM helices that line the lateral openings. Extracellular and cytoplasmic views are also shown.

positions that provide good coverage of the conformational change predicted by the models. We then introduced cysteine residues at these positions on a starting construct in which the one native cysteine in LtaA was replaced with serine. The cysteine-less LtaA variant effectively performed glycolipid flipping in proteoliposomes (Supplementary Fig. 2a–c). The three mutants F45C-T253C, A53C-T366C, and K166C-I250C were then irreversibly cross-linked with N,N'-(o-phenylene)-dimaleimide (o-PDM), which has a spacer arm length of 6 Å. Cross-linked and non-cross-linked LtaA mutants were digested with either trypsin or chymotrypsin, and analyzed by high-resolution liquid chromatography–mass spectrometry (LC–MS) to evaluate the presence of non-cross-linked cysteine-containing peptides. The peptides abundance was normalized against an internal reference peptide. We successfully identified non-cross-linked peptides in untreated samples of the three mutants: F45C-T253C, A53C-T366C, and K166C-I250C (Fig. 2a and Supplementary Fig. 3). The abundance of these peptides was clearly diminished in the cross-linked protein samples (Fig. 2a), demonstrating that the selected pairs of residues are in proximity as predicted in the inward-facing models.

As a control, we performed a similar experiment but with pairs of residues that were shown to interact at the cytoplasmic region of the outward-facing structure (Fig. 2b). Thus, we introduced cysteine residues at the positions K80-E339 and K141-N276, present at the cytoplasmic ends of TM2-TM11 and TM5-TM8, respectively. Cβ-Cβ distances between these residues are smaller than 8 Å in the outward-facing structure, but larger than 12 Å in the inward-facing models (Supplementary Fig. 1). LC–MS analysis of the double mutants K80C-E339C and K141C-N276C confirmed the proximity of these residues as non-cross-linked peptides are more abundant in untreated samples, whereas in the presence of the cross-linking agent their abundances decrease substantially (Fig. 2b). In summary, our cross-linking analysis supports the predicted conformations and interactions reported by the inward-facing models of LtaA and indicate the position of residues to guide cysteine disulfide trapping of LtaA conformations.

**Alternating conformations in proteoliposome membranes**. We investigated the conformations displayed by LtaA in membranes by evaluating the cross-linking of double-cysteine mutants reconstituted in proteoliposomes (Fig. 2c). The cysteine pairs reported on the conformation of the TM helices that line the lateral openings, TM2-TM11 and TM5-TM8 (Fig. 1c, d). We screened for successful cross-links by using a gel-shift assay in which we first incubated with the o-PDM cross-linker, followed by treating the proteoliposomes with 5-kDa PEG-maleimide (mPEG5k)[50]. This treatment generates a substantial shift in the protein mobility in polyacrylamide gel electrophoresis as mPEG5k irreversibly binds free cysteines. However, if the introduced cysteines are cross-linked by o-PDM, then they will not react with mPEG5k and no shift in gel mobility would be observed. A band indicating LtaA dimer was frequently observed in gels for all variants, including the cysteine-less LtaA (Fig. 2c and Supplementary Fig. 4c). However, since LtaA was shown to be monomeric after purification and reconstitution in nanodiscs (Supplementary Fig. 2d), we consider the dimer band to be an in-gel artifact.

We evaluated the cross-linking of residues A53C-T366C (TM2-TM11) and K166C-I250C (TM5-TM8), positioned at the extracellular region, and K80C-E339C (TM2-TM11) and K141C-N276C (TM5-TM8), located at the cytoplasmic region (Fig. 2c and Supplementary Fig. 4a–c). Before cross-linking, each double-cysteine mutant showed a gel shift after incubation with mPEG5K, thus demonstrating PEGylation of free cysteines (Fig. 2c and Supplementary Fig. 4c). In contrast, after treatment with o-PDM, all the double-cysteine mutants were protected from PEGylation, thus showing that all mutants were successfully cross-linked. The cysteine-less control LtaA, did not show a gel shift in any of the conditions (Fig. 2c and Supplementary Fig. 4c), demonstrating that the shifts observed for the mutants were due to PEGylation of cysteines. These results support that when embedded in the membrane of proteoliposomes, LtaA can adopt conformations where residues at the lateral openings lined by TM2-TM11 and TM5-TM8 display similar distances to those reported by the outward-facing structure and the inward-facing models.

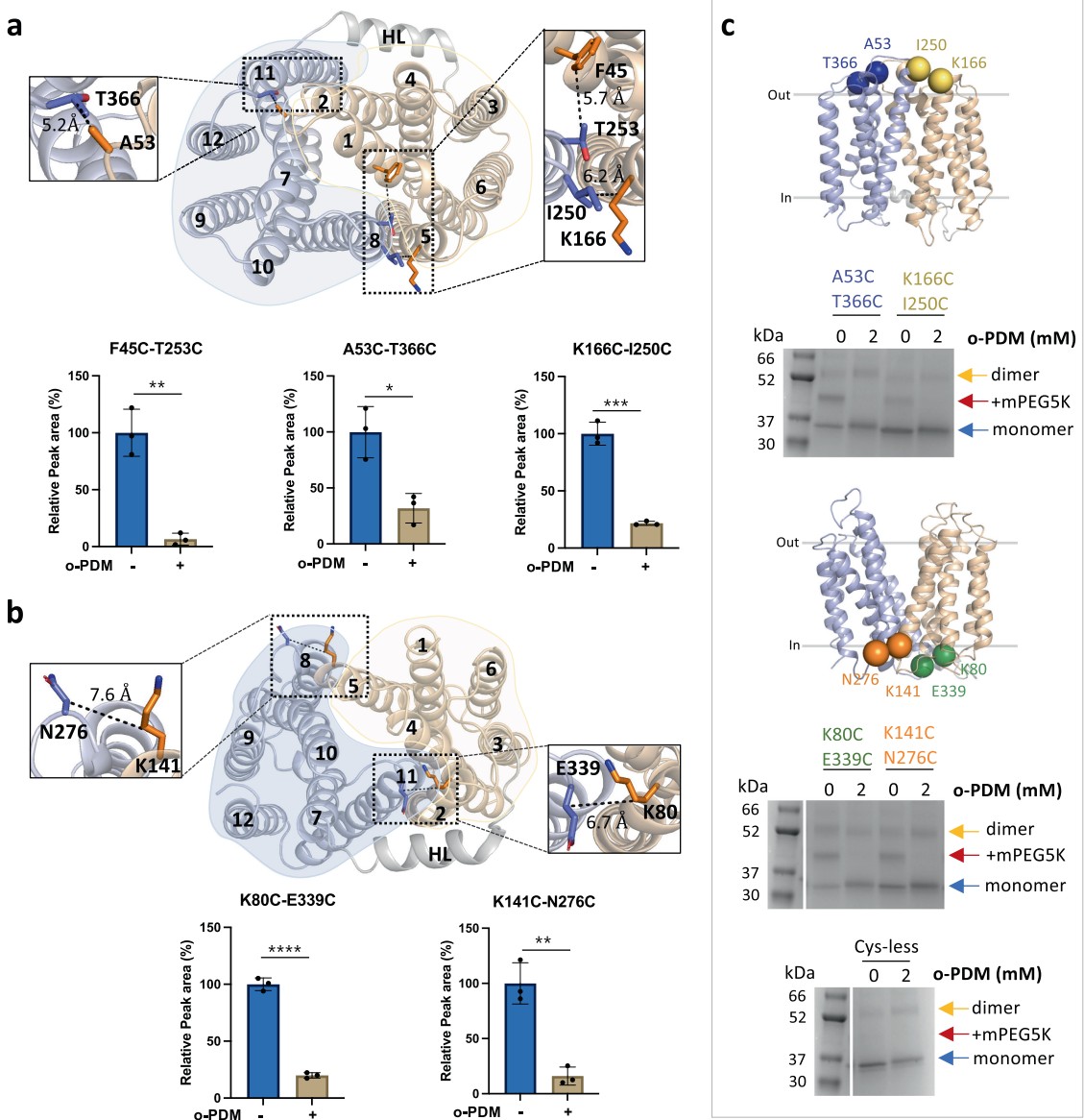

**Fig. 2 LtaA adopts inward- and outward-facing states.** Selected residues for cross-linking of LtaA in inward-facing conformation (**a**) and outward-facing conformation (**b**). N-terminal and C-terminal domains are shown in light orange and light blue, respectively. The relative abundance of cysteine-containing peptides in absence (−) or presence (+) of N,N′-(o-phenylene)-dimaleimide (o-PDM) is shown in histograms. Collision-induced dissociation (CID) spectrum of cysteine-containing peptides and elution profiles of peptide fragments are shown in Supplementary Fig. 3. Error bars indicate +/− standard deviation (s.d.) ($n = 3$, biological replicates). *$P \leq 0.05$, **$P \leq 0.01$, ***$P \leq 0.001$, ****$P \leq 0.0001$. Asterisks mark the result from unpaired $t$ test. **c** Cross-linking analysis of LtaA in proteoliposomes. Positions of selected cysteine pairs at the extracellular and cytoplasmic regions of LtaA are shown as spheres. SDS-PAGE show band shifts of samples treated with mPEG5K after irreversible cross-link with o-PDM. Separated species are indicated with arrows. The complete gel is shown in Supplementary Fig. 4C. SDS-PAGE experiments were independently repeated at least three times with similar results. Source data are provided as a Source Data file.

**Dynamics of LtaA alternating conformations in membranes.** For lipid transporters that adopt a "trap-and-flip" mechanism, substrate binding and release involve the movement of lipids through lateral openings of the translocation channel[18,21,22,41,44,45]. We studied the dynamic behavior of the lateral openings in different conformational states of LtaA when the protein is embedded in a lipid bilayer. To do this, we performed molecular dynamics (MD) simulations of outward- and inward-facing LtaA in a membrane composed of POPG (65%), diacylglycerol (20%), cardiolipin (10%), and gentiobiosyl-diacylglycerol (5%), resembling the membrane of *S. aureus*[51]. During the simulations, outward- and inward-facing states were found to be stable as judged by RMSD and RMSF plots (Supplementary Fig. 5a–e). The

simulations revealed that all the optimized inward-facing models and the AF model exhibit a cavity which is open to the cytoplasm and closed to the extracellular space, whereas the cavity of the outward-facing state is open to the extracellular space and closed to the cytoplasm (Fig. 3a, b). During the simulations, the AF inward-facing model displayed wide lateral openings, making the central translocation pathway accessible to the surrounding membrane and solvent (Fig. 3a, b). In contrast, the central translocation pathway in the "repeat-swap" inward-facing models was less accessible due to their narrower lateral openings (Fig. 3a, b).

In agreement with the observed wider opening of the extracellular lateral openings, the simulations of outward-facing

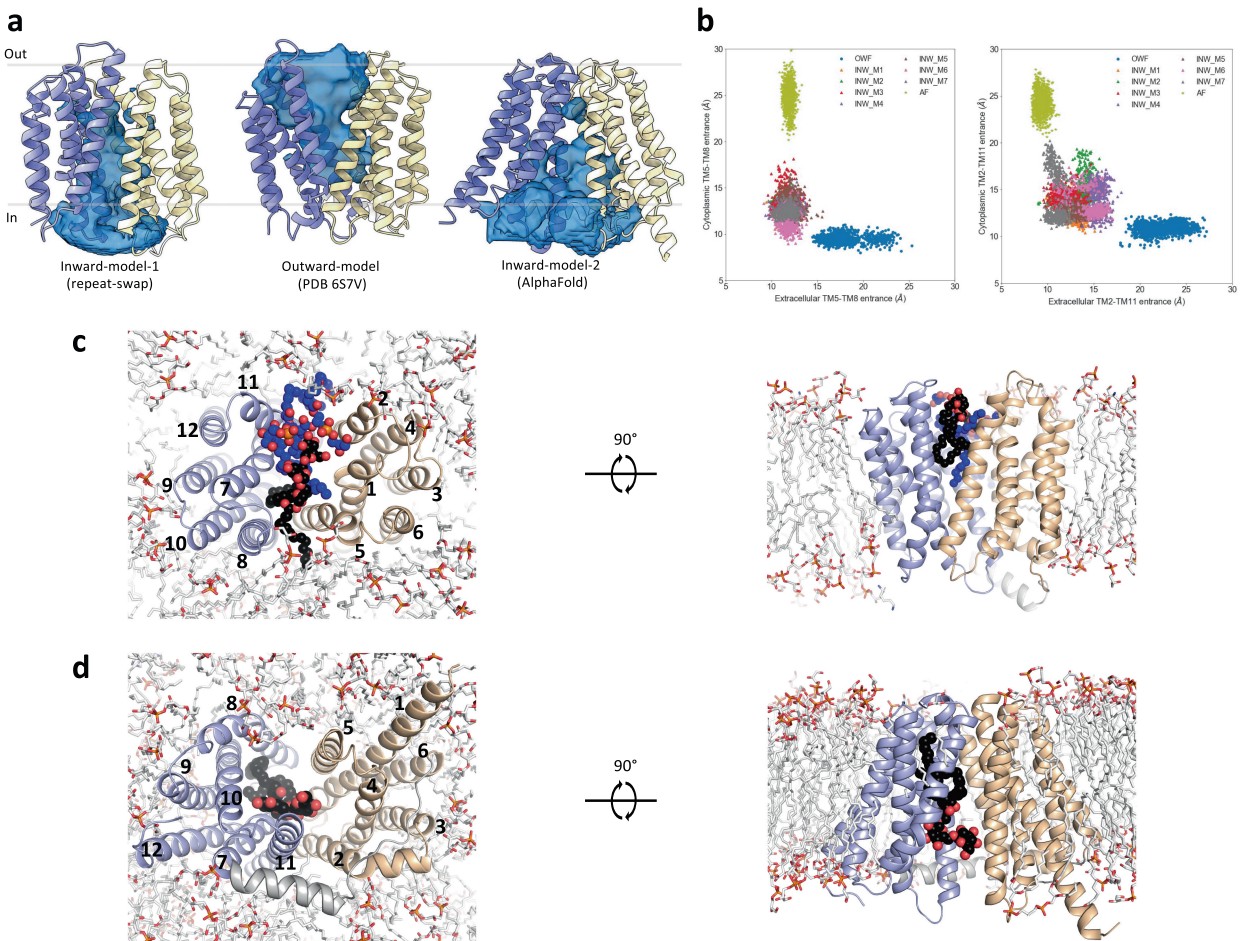

**Fig. 3 Lateral openings facilitate access of glycolipids to the central translocation pathway. a** Representative view of the solvent-exposed cavity of inward-facing and outward-facing LtaA as observed during MD simulations. **b** Analysis of distances between TM helices lining the cytoplasmic and extracellular lateral openings of outward-facing and inward-facing models. The center of masses of the Cα atoms of extracellular residues 52–57 (TM2), 163–167 (TM5), 250–255 (TM8), 364–367 (TM11), and of cytoplasmic residues 77–81 (TM2), 139–143 (TM5), 273–276 (TM8), 341–344 (TM11), were used for the calculation of inter-TM distances. **c** Intrusion of gentiobiosyl-diacylglycerol (black spheres) and POPG (blue spheres) molecules in the extracellular cavity of LtaA during simulations. **d** Intrusion of gentiobiosyl-diacylglycerol (black spheres) in the intracellular cavity of the AF inward-facing model of LtaA during simulations. N-terminal and C-terminal domains are shown in light orange and light blue, respectively.

LtaA showed the intrusion of glycolipid and POPG molecules into the putative translocation pathway (Fig. 3b, c and Supplementary Movie 1). The glycolipid was seen to intrude from the TM5-TM8 opening, with one of the aliphatic tails reaching to the C-terminal hydrophobic pocket, whereas two POPG molecules intrude from the TM2-TM11 opening (Fig. 3c and Supplementary Movie 1). In a similar manner, the wide intracellular lateral openings of the AF inward-facing model showed the intrusion of a glycolipid into the putative translocation pathway from the side of the TM5-TM8 opening (Fig. 3d and Supplementary Movie 2), with the aliphatic tails reaching to the C-terminal hydrophobic pocket. No glycolipid was observed intruding from the TM2-TM11 opening, mainly because of the obstruction by the horizontal helix (Fig. 3d and Supplementary Movie 2). Taking together, these results support that binding and release of the glycolipid by LtaA, involves movement of the substrate through lateral openings, which grant direct access to the surrounding bilayer.

**Alternating access to the central cavity is essential for function**. The cross-linking results described above showed that LtaA could cycle through outward- and inward-facing conformations. In addition, MD simulations suggest that lateral openings allow the

passage of lipids into and out of the central cavity. However, the functional relevance of the lateral openings and cycling through alternating conformations is unknown. Understanding this is important because some flippases and scramblases use a "credit card" mode of transport, where exposing a side cleft or a cavity to one side of the membrane is sufficient for catalysis of lipid transport across the membrane[37–39,42,52]. Thus, we performed copper chloride catalyzed cross-linking of residues located at the lateral openings lined by TM2-TM11 and TM5-TM8 and then determined proton-coupled glycolipid transport activity of cross-linked LtaA variants in proteoliposomes (Fig. 4a–d and Supplementary Fig. 4d). In this assay, the addition of the K$^+$-selective ionophore valinomycin generates a membrane potential of about −60 mV, which drives proton influx. Acidification of the lumen of proteoliposomes quenches the fluorophore 9-amino-6-chloro-2-methoxyacridine (ACMA) causing a decrease in the fluorescence[27]. The double-cysteine mutants A53C-T366C (TM2-TM11) and K166C-I250C (TM5-TM8) close the extracellular side openings, whereas the mutants K80C-E339C (TM2-TM11) and K141C-N276C (TM5-TM8) close the cytoplasmic openings (Fig. 4a, b and Supplementary Fig. 4a, b). Our results show that individual cross-linking of the lateral openings decreases LtaA activity relative to non-cross-linked samples (Fig. 4a, b). The

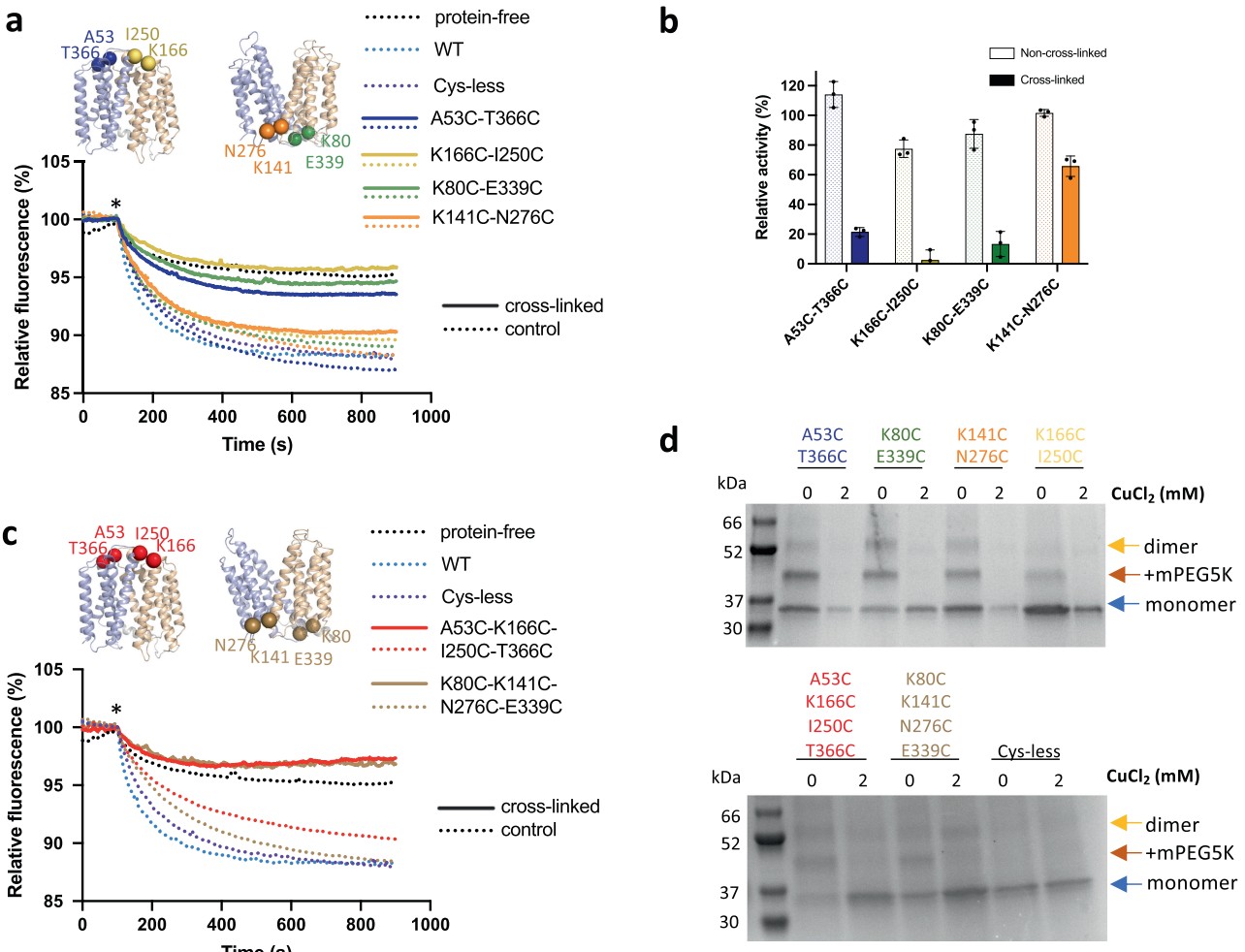

**Fig. 4 Cycling through outward- and inward-facing conformations is essential for LtaA activity. a**, **c** Proton-transport activity of LtaA and variants after chemical cross-linking with $CuCl_2$ (solid lines) or in absence of cross-linking treatment (dotted lines) ($n \geq 3$). Proteoliposomes and protein-free liposomes containing 100 mM KCl were diluted in buffer containing 10 mM KCl, 90 mM NaCl, and ACMA. $H^+$ influx was initiated by establishing a membrane potential upon the addition of the potassium ionophore valinomycin (asterisk). **b** Relative activity of cross-linked and non-cross-linked LtaA variants measured in A. Relative activity $= 100 \times (F'_i - F'_{liposomes})/(F'_{wt} - F'_{liposomes})$, where i corresponds to each variant, liposomes correspond to protein-free liposomes, and F′ correspond to the relative fluorescence at the plateau (800 s). Error bars show $+/-$ s.d. of technical replicates, $n = 3$. **d** SDS-PAGE shows band shifts of proteoliposome samples treated with mPEG5K after cross-linking with $CuCl_2$. Separated species are indicated with arrows. SDS-PAGE experiments were independently repeated at least three times with similar results. Source data are provided as a Source Data file.

activity of cross-linked samples of the intracellular side opening lined by TM2-TM11 and that of both extracellular openings was observed to decrease prominently, whereas cross-linking of the lateral opening lined by TM5-TM8 reduces LtaA activity to about two-thirds of its level (Fig. 4a, b).

In addition, we aimed to completely close the cytoplasmic or extracellular cavities and test the effect on LtaA activity (Fig. 4c). To do this, we constructed the mutant A53C-T366C-K166C-I250C that after cross-linking would close the extracellular pathway, while the mutant K80C-E339C-K141C-N276C would close the cytoplasmic pathway (Fig. 4 and Supplementary Fig. 4a, b). Our results show that, in contrast to non-cross-linked proteins, both mutants display background quenching levels, similar to that observed for protein-free liposomes, thus, indicating strong inhibition of transport activity (Fig. 4c). We confirmed cross-linking of each double- and tetra-cysteine mutant reconstituted in proteoliposomes by gel-shift assays after incubation with mPEG5K (Fig. 4d and Supplementary Fig. 4d), which showed that after treatment with copper chloride, all the mutants were protected from PEGylation, whereas before cross-

linking a gel shift was observed. This confirmed that all mutants were successfully cross-linked in the proteoliposomes samples used in the assay.

Taking together, these results reveal that alternating opening to both sides of the membrane is a requirement for LtaA function. However, not all lateral openings seem to have the same functional relevance. In particular, our results demonstrate that while both extracellular lateral openings are similarly important for function, the cytoplasmic lateral opening lined by TM2 and TM11 has a more significant role, as revealed by the low activity of the cross-linked variant K80C-E339C. By contrast, cross-linking of the cytoplasmic lateral opening lined by TM5 and TM8, K141C-N276C, affect LtaA function less strongly.

**The hydrophobic pocket is relevant for lipid transport.** Inspection of the central cavity shows that similar to what was observed in the outward-facing crystal structure of LtaA[27], the central cavity of the inward-facing models is amphipathic (Fig. 5a). The cavity displays a hydrophilic pocket, enclosed

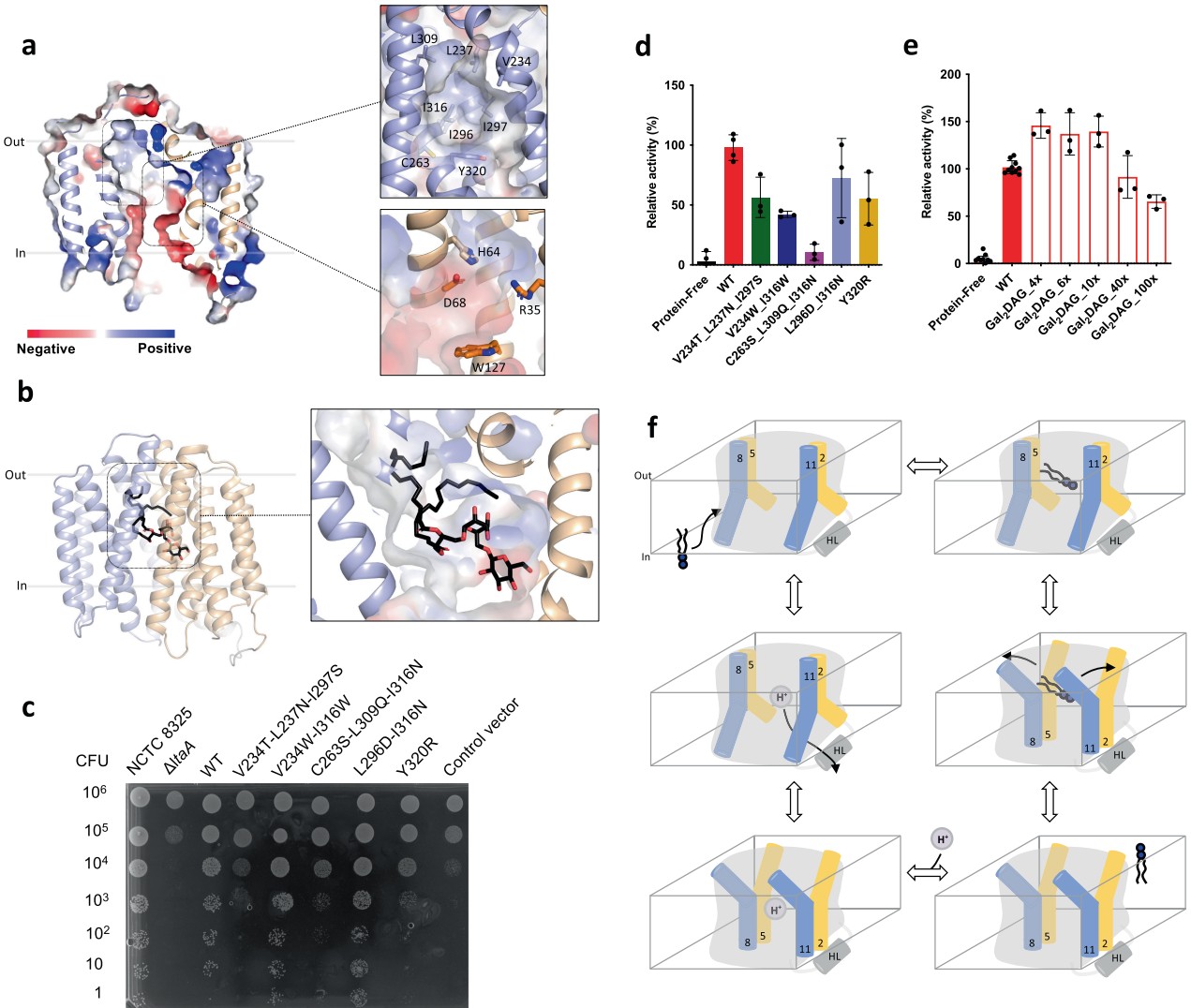

**Fig. 5 Hydrophilic and hydrophobic cavities participate in "trap-and-flip" of lipids. a** Vacuum electrostatic surface representation of inward-facing model of LtaA. Residues forming the hydrophobic and hydrophilic pockets are shown. **b** A model of a glycolipid molecule docked into the amphipathic cavity of LtaA. The lipid tail length corresponds to C16 chains. N-terminal and C-terminal domains are shown in light orange and light blue, respectively. **c** *S. aureus* cell growth on LB agar plates containing 0.1 mM IPTG, buffered at pH 6.4. The *ΔltaA* mutant is complemented with pLOW vector carrying a *ltaA*-WT gene or the annotated point mutations; Control vector indicates the pLOW vector carrying an unrelated gene (dCas9). **d** Mutagenesis analysis of the hydrophobic pocket. Relative flipping activity of LtaA-WT and variants. Error bars show $+/-$ s.d. of technical replicates, $n = 3$. **e** Headgroup selectivity analysis. Relative flipping activity of LtaA in the presence of different concentrations of digalactosyldiacylglycerol (Gal$_2$DAG). Molar excess of Gal$_2$DAG over Glc$_2$-DAG-NBD is indicated. Error bars show $+/-$ s.d. of technical replicates, $n \geq 3$. Source data are provided as a Source Data file. **f** Proposed mechanism of LtaA catalyzed glycolipid transport. Schematic of conformational states throughout LtaA transport cycle.

mainly by residues from the N-terminal domain (E32, R35, D68, W127, and W150), which we have previously shown to be relevant for recognition of the glycolipid headgroup and proton transport[27], and a hydrophobic pocket, enclosed mainly by residues from the C-terminal domain (V234, L237, C263, L296, L300, L309, I316, and Y320) (Fig. 5a). The recent structure of the MFSD2A transporter, trapped in an inward-facing conformation, displays a similar amphipathic central cavity (Supplementary Fig. 6)[21,22]. MD simulations and computational docking of a glycolipid molecule to inward-facing LtaA suggest that the gentiobiosyl headgroup is preferentially accommodated in the hydrophilic pocket, whereas the diacylglycerol aliphatic tails are docked into the hydrophobic pocket (Figs. 3d and 5b).

A striking feature of the central cavity observed in LtaA and MFSD2A[21,22,27], and to our knowledge, not observed in other

MFS structures available to date, is the presence of the highly hydrophobic pocket at the C-terminal domains of these transporters (Fig. 5a and Supplementary Fig. 6). To test the importance of this pocket in LtaA, we have designed mutants that introduce polar residues, thus making it more hydrophilic. We then evaluated the growth of *S. aureus* NCTC8325 *ΔltaA* cells complemented with ectopic copies of the *ltaA* gene carrying these mutations (Fig. 5c and Supplementary Fig 7). The variants V234T/L237N/I297S, C263S/L309Q/I316N, and Y320R display marked growth defects, whereas the mutant L296D/I316N do not affect growth. Each mutant was also purified and reconstituted into proteoliposomes, followed by determination of their flipping activity (Fig. 5d). In agreement with the results from *S. aureus* growth assays, the mutants V234T/L237N/I297S, C263S/L309Q/I316N, and Y320R display low relative activity compared to LtaA-

WT (Fig. 5d), whereas L296D/I316N display the highest activity among all mutants. In contrast, introducing a mutation that scarcely increases the polarity of the cavity but that changes the size of residues V234 and I316, displayed low relative flipping activity compared to LtaA-WT, but does not affect the growth of *S. aureus* NCTC8325 *ΔltaA* cells, likely due to the remaining activity of this mutant to suffice for lipoteichoic acid synthesis under in vivo conditions (Fig. 5c, d and Supplementary Fig 7). Taken together, these results support a fundamental role of the hydrophobic pocket in glycolipid transport. As suggested by MD simulations and docking analysis, it is likely that this pocket is involved in binding of the aliphatic tails of the glycolipid substrate. The striking hydrophobicity of the C-terminal TM helices 7, 8, and 10 in multiple MFS lipid transporters (Supplementary Fig. 8), and the involvement in coordination of the aliphatic chain of lysophospholipid as revealed by the structure of MFSD2A[21,22], suggest a shared mechanistic role of the hydrophobic pocket in lipid-tails binding in MFS lipid transporters.

**The hydrophilic pocket dictates substrate specificity.** So far, our results suggest that during transport, LtaA encloses the full glycolipid substrate in the amphipathic cavity. However, understanding the relevance of the individual parts of the substrate molecule, headgroup and aliphatic chains, is fundamental for the future design of pharmacologically relevant molecules targeting this and other MFS lipid transporters. To gain insight into whether LtaA displays higher selectivity toward the headgroup than for the diacylglycerol moiety, we performed flipping assays with LtaA-WT co-reconstituted in proteoliposomes together with NBD-labeled Glc$_2$-DAG (gentiobiosyl-diacylglycerol) and increasing concentrations of Gal$_2$-DAG (digalactosyldiacylglycerol) (Fig. 5e). Glucose and galactose differ only in the orientation of the –OH group at the C-4 position. Thus, we hypothesized that if the headgroup is more relevant for substrate recognition than the aliphatic chains, then transport of Glc$_2$-DAG-NBD will not be affected, since the difference between glucose and galactose would prevent Gal$_2$-DAG from being a good competitor. On the other hand, if the diacylglycerol moiety is more relevant for substrate recognition, we expect Gal$_2$-DAG to be a strong competitor, thus resulting in a marked decrease in Glc$_2$-DAG-NBD transport. Our results show that even under a high excess of Gal$_2$-DAG, there is no significant effect on Glc$_2$-DAG-NBD transport (Fig. 5e). We have previously shown that gentiobiose (β-D-Glc-(1,6)-D-Glc), a disaccharide with the same composition and conformation as the glycolipid headgroup (Glc$_2$-DAG), inhibits lipid transport[27]. Taken together, these results suggest that an intact headgroup is highly relevant for substrate binding and transport, and that even minor changes to the headgroup abolish recognition. Independent of the presence of the diacylglycerol moiety and its predicted binding to the hydrophobic pocket, the headgroup seems to dictate whether a glycolipid can be a substrate for LtaA or not.

## Discussion

Several transporters of the MFS superfamily have been structurally characterized in one or multiple conformational states[4,8–17]. However, except for the outward-facing structure of LtaA[27], solved by X-ray crystallography, and the inward- and outward-facing structures of MFSD2A[21,22], solved by single-particle cryo-electron microscopy, there are no additional structures available of MFS lipid transporters. Despite the differences among their lipid substrates, the distinct composition of bacterial and eukaryotic membranes, and their opposite vectorial lipid transport directions, LtaA and MFSD2A share multiple architectural similarities, including a canonical MFS fold of 12 TM helices and an amphipathic central cavity with asymmetric distribution of hydrophobic and hydrophilic residues (Supplementary Fig. 6). A similar arrangement of central cavity residues has been predicted to be present in the bacterial lysophospholipid transporter LplT[18], and are likely to be part of the architecture of other MFS lipid transporters (Supplementary Fig. 8). These characteristics suggest a common mechanism of substrate recognition and translocation among these proteins. Indeed, LtaA and MFSD2A display strong selectivity towards the headgroup of their lipid substrates[6,27]. In the case of MFSD2A, the zwitterionic charge of the phosphatidylcholine headgroup is fundamental for ligand transport, whereas LtaA displays strong selectivity towards the gentiobiosyl disaccharide headgroup of the glycolipid. Furthermore, LtaA selects against an isomer of the disaccharide headgroup, as shown by the poor competition displayed by digalactosyldiacylglycerol in transport assays (Fig. 5e). In contrast, LplT has been shown to exhibit a more relaxed specificity towards the lipid headgroup, being able to transport lysophosphatidylethanolamine and lysophosphatidylglycerol lipids[18].

Although MFSD2A and LplT have been shown to strongly select for lysophospholipids, they display relaxed selectivity towards the length of the aliphatic chains[6,18]. MFSD2A transports docosahexaenoic acid (DHA), an essential omega-3 fatty acid for brain growth and cognitive function, in the form of lysophosphatidylcholine, but can also transport other lipids with at least 14-carbons acyl chain[6]. It is noteworthy that *S. aureus* membranes are rich in diacylglycerols with chains length ranging from C$_{15}$ to C$_{18}$, with the most dominant lipid species having a C$_{18}$:C$_{15}$ composition[26]. This variability among diacylglycerols in *S. aureus*, and the measurable translocation of Glc$_2$-DAG-NBD[27], which has a C$_{10}$ acyl chain length and an NBD group linked to one of the diacylglycerol chains, suggest that LtaA displays similar relaxed specificity toward the length of the lipid part.

Our results strongly suggest that in contrast to mechanisms proposed for other lipid transporters, LtaA transports gentiobiosyl-diacylglycerol by a "trap-and-flip" mechanism, which follows the classical alternating-access model of transport[53], with the entire glycolipid entering and leaving the central translocation pathway (Fig. 5f). The transition between inward- and outward-facing states likely follows the "rocker-switch" alternating-access model that describes the mechanism of MFS transporters of water-soluble substrates[4,54,55]. In this model, the N-terminal and the C-terminal domains rotate about an axis crossing the center of the transporter. This "rocking" motion facilitates switching between the two conformations[4,54,55]. However, whereas the two extracellular lateral openings display similar widths and dynamics during MD simulations (Fig. 3b) and similar behavior in transport assays (Fig. 4a, b), the cytoplasmic lateral openings seem to display specialized functions during the transport cycle of LtaA (Fig. 4a, b). Entry of the glycolipid into the putative translocation pathway from the side of the TM5-TM8 opening is supported by MD simulations of the AF inward-facing model (Fig. 3d and Supplementary Movie 2), albeit entry through the opening lined by TM2-TM11 might also be possible since cross-linking of the TM5-TM8 opening (K141C-N276C) decreases LtaA activity moderately. In contrast, cross-linking of the lateral opening lined by TM2-TM11 decreases LtaA activity significantly (Fig. 4a, b). Since multiple residues located in TM2 likely participate in proton transport as shown before[27], we speculate that "gating" of the TM2-TM11 opening is essential for proton coupling. A similar role of charged residues in TM2 involved in ion-dependent gating has been postulated for MFSD2A and supported by MD simulations[22]. Thus, we propose that during its transport cycle (Fig. 5f), inward-facing LtaA binds a glycolipid molecule which enters through the lateral opening

lined by TM5 and TM8, triggering the conformational change to outward-facing conformations, in which the glycolipid is released into the membrane presumably through any of the two extracellular lateral openings. Protonation of residues in the hydrophilic pocket allows transition to inward-facing conformations, followed by proton release to the cytoplasm through the lateral opening lined by TM2 and TM11 (Fig. 5f).

In summary, our results provide insights into the molecular mechanism of glycolipid transport by LtaA and support a "trap-and-flip" model where lateral "gates" display distinct mechanistic roles. Our data suggest that the highly selective hydrophilic pocket dictates substrate specificity, but that the hydrophobic pocket is fundamental for aliphatic chains' transport. The mechanistic elements described here might be shared by other MFS lipid transporters and can be decisive for the design of drugs targeting these proteins.

## Methods

**LtaA expression and purification**. The gene encoding *S. aureus* LtaA was cloned into a modified pET-19b vector (Novagen), with an N-terminal His$_{10}$ affinity tag. LtaA-WT and mutants were expressed in *E. coli* BL-21 Gold (DE3) (Stratagene) cells. Cells were grown in Terrific Broth (TB) medium supplemented with 1% glucose (w/v) at 37 °C. Overexpression was induced with 0.2 mM Isopropyl β-D-1-thiogalactopyranoside (IPTG) for 1 h. All following steps were performed at 4 °C, unless differently specified. Cells were harvested by centrifugation, resuspended in 50 mM Tris-HCl, pH 8.0; 500 mM NaCl; 5 mM β-mercaptoethanol; 0.5 mM PMSF and disrupted in a M-110L microfluidizer (Microfluidics) at 10000 psi chamber pressure. Membranes were pelleted by ultracentrifugation and solubilized in 50 mM Tris-HCl, pH 8.0; 200 mM NaCl; 20 mM imidazole; 15% glycerol (v/v); 5 mM β-mercaptoethanol; 1% lauryl maltose neopentyl glycol (LMNG, Anatrace); 1% N-dodecyl-β-D-maltopyranoside (w/v) (DDM, Anatrace). After removing debris, the supernatant was loaded onto a pre-equilibrated NiNTA superflow affinity column (Qiagen). The column was washed with 50 mM Tris-HCl, pH 8.0; 200 mM NaCl; 50 mM Imidazole; 10% glycerol (v/v); 5 mM β-mercaptoethanol; 0.02% LMNG and 0.02% DDM and then further washed with the same buffer only containing 0.02% LMNG. Elution was performed in the same buffer containing 200 mM Imidazole. Buffer exchange to buffer 10 mM Tris-HCl pH 8.0; 150 mM NaCl; 0.02% LMNG; with or without 2 mM β-mercaptoethanol, was performed using PD-10 columns (GE Healthcare). If necessary, analytical size-exclusion chromatography was performed on a Superdex 10/300 GL column (GE Healthcare) in buffer 10 mM Tris-HCl, pH 8.0; 150 mM NaCl; 0.02% LMNG[56].

**Mutagenesis**. LtaA mutants were generated using overlap Extension-PCR, followed by DpnI digestion for two hours at 37 °C, and transformation into *E. coli* DH5α cells. The mutations were confirmed by DNA sequencing (Microsynth). All oligos used for mutagenesis are listed in Supplementary Table 2.

**YpfP expression and purification**. The gene encoding *S. aureus* YpfP was cloned into a modified pET-19b vector (Novagen) with an N-terminal His$_{10}$ affinity tag. YpfP was overexpressed in BL-21 Gold (DE3) (Stratagene) cells. Cells were grown in TB medium supplemented with 1% glucose (w/v) at 37 °C until a cell density of OD$_{600}$ = 3. Subsequently, cells were induced with 0.2 mM IPTG for 16 h at 24 °C. Cells were harvested by centrifugation and resuspended in buffer A (50 mM Tris-HCl pH 8.0; 200 mM NaCl; 3% glycerol; 3 mM β-mercaptoethanol) plus 0.5 mM PMSF. Cells were disrupted using a tip sonication. After differential centrifugation, the supernatant containing YpfP was incubated with NiNTA resin and left stirring for 1 h at 4 °C. Washing was performed with buffer A complemented with 50 mM imidazole pH 8.0, followed by elution with buffer A complemented with 200 mM imidazole pH 8.0. YpfP was desalted in buffer 50 mM Tris-HCl pH 8.0; 200 mM NaCl; 10% glycerol using PD-10 columns (GE healthcare). If required YpfP was concentrated using a Vivaspin 20 30MWCO until 2.4 mg/ml, flash-frozen in liquid nitrogen, and stored at −80 °C until further use.

**Synthesis of NBD-glycolipid and glycolipid**. Synthesis of glycolipd and nitrobenzoxadiazole (NBD)-labeled glycolipid was performed using a modification of the protocol described by Jorasch et al.[57] and Kiriukhin et al.[58]. A final concentration of 2 mM UDP-Glucose (Sigma), 2 mM NBD-decanoyl-2-decanoyl-sn-Glycerol (Cayman), and 1.2 mg/ml purified YpfP were incubated together for 16 h at 30 °C. The reaction product was separated using thin-layer chromatography (TLC) with a silica gel matrix (Sigma) in a solvent mixture consisting of chloroform:methanol:water (65:25:4, vol/vol/vol). Silica containing the NBD-glycolipid was recovered from plates, and the NBD-glycolipid was extracted from the silica by incubation with a solvent mixture of chloroform:methanol (50:50, vol/vol), followed by drying of the anchor-LLD under argon atmosphere, and subsequently resuspension in 20 mM Tris-HCl pH 8.0; 150 mM NaCl. NBD-glycolipid was flash-

frozen in liquid nitrogen, and stored at −80 °C until further use. Reaction products were previously characterized[27]. Non-labeled glycolipid was prepared similarly by incubation of 2 mM UDP-Glucose, 2 mM 1,2-dimyristoyl-sn-glycerol (Avanti), and 1.2 mg/ml YpfP for 16 h at 30 °C.

**Formation of LtaA proteoliposomes**. LtaA was reconstituted in unilamellar liposomes prepared by extrusion through polycarbonate filters (400-nm pore size) from a 3:1 (w/w) mixture of *E. coli* polar lipids and L-α-phosphatidylcholine (Avanti polar lipids) resuspended in 20 mM Tris-HCl pH 8.0; 150 mM NaCl, and 2 mM β-mercaptoethanol. After saturation with DDM (Anatrace), liposomes were mixed with purified LtaA in a 50:1 (w/w) lipids/protein ratio. DDM was removed after incubation with BioBeads (BioRad). Proteoliposomes were centrifugated, washed, and resuspended to a final concentration of 20 mg/ml lipids; 7.8 µM LtaA. The proteoliposomes were flash-frozen in liquid nitrogen and stored at −80 °C until further use.

**In vitro flipping assay**. Before performing flipping assays, proteoliposomes were thawed, their resuspension buffer was exchanged to 20 mM MES pH 6.5; 150 mM NaCl, and the product of the NBD-glycolipid synthesis reaction was incorporated by performing freeze/thaw cycles. Proteoliposomes and protein-free liposomes were diluted to a concentration of 2 mg/ml lipids followed by extrusion through polycarbonate filters (400-nm pore size). Proteoliposomes were immediately used for flipping assays. In case of competition assays with digalactosyldiacylglycerol (DGDG). DGDG powder (Avanti) was resuspended in 20 mM Tris-HCl; 150 mM NaCl and incorporated into proteoliposomes during freeze/thaw cycles together with the NBD-glycolipid. Flipping of NBD-glycolipid was assessed by determining the percentage of NBD-fluorescence that is quenched after the addition of a 5 mM sodium dithionite (Sigma) after 200 seconds of starting fluorescence recording. Hundred seconds before finishing data recording, 0.5% Triton X100 was added to permeabilize the liposomes, making all NBD-glycolipid molecules accessible to dithionite reduction. The fluorescence after Triton X100 addition was used for baseline calculations. Fluorescence was recorded at 20 °C using a Jasco Fluorimeter. The excitation and emission wavelengths were 470 and 535 nm, respectively. For analysis, the fluorescence intensity was normalized to F/F$_{max}$. Relative flipping activities were calculated as follows: relative activity = 100 × ((F/F$_{max}$)$_i$ − (F/F$_{max}$)$_{liposomes}$)/((F/F$_{max}$)$_{wt}$ − (F/F$_{max}$)$_{liposomes}$), where i corresponds to each respective treatment/mutants, liposomes corresponds to liposomes without protein, wt corresponds to wild-type LtaA and F/Fmax values correspond to the normalized fluorescence values at the plateau after addition of sodium dithionite. Curves were plotted using GraphPad Prism 8. Time courses of the dithionite-induced fluorescence decay in liposomes were repeated at least three times for each individual experiment.

**Proton-transport assay**. LtaA proteoliposomes and protein-free liposomes were thawed, and their internal buffer was exchanged to 5 mM HEPES pH 7.3; 100 mM KCl. Glycolipid was incorporated during freeze/thaw cycles followed by extrusion through polycarbonate filters (400-nm pore size). After 90 s of sonication, proteoliposomes and protein-free liposomes were diluted 25-fold in buffer containing 5 mM HEPES pH 7.3; 10 mM KCl; 90 mM NaCl; 0.5 µM 9-amino-6-chloro-2-methoxyacridine (ACMA). Fluorescence was recorded using a Jasco Fluorimeter with excitation and emission wavelengths of 410 and 480 nm, respectively. When the fluorescence signal was stable, H$^+$ influx was initiated by establishing a membrane potential by the addition of the potassium ionophore valinomycin (5 nM). Time courses of the proton-transport assay in proteoliposomes were repeated at least three times for each individual experiment. Cross-linking was performed before the measurement by the addition of 2 mM CuCl$_2$ to the proteoliposomes during the buffer exchange and incorporation of glycolipid steps. After 1 h incubation at RT in the dark, CuCl$_2$ was removed by centrifugation, and proteoliposomes were resuspended in buffer 5 mM HEPES, pH 7.3; 100 mM KCl.

**LtaA cross-linking and PEGylation**. Before performing cross-linking, proteoliposomes were thawed, their resuspension buffer was exchanged to 20 mM Tris-HCl pH 8.0; 150 mM NaCl LtaA mutants incorporated into proteoliposomes or in detergent micelles were incubated with 2 mM CuCl$_2$ or N,N'-1,2-phenylenedimaleimide (o-PDM) for 30 min to 1 h at RT in the dark. In the case of non-cross-linked samples, proteoliposomes were incubated with a proportional volume of DMSO or buffer. Crosslinkers were removed by centrifugation and washing with buffer or by buffer exchange to 20 mM Tris-HCl, pH 8.0; 150 mM NaCl using Zeba™ Spin Desalting Columns, 7 K MWCO, 0.5 mL (ThermoScientific). To PEGylate free cysteines, LtaA mutants were incubated for 3 h at RT in the presence of 0.5 mM mPEG5K-Maleimide (Sigma) and 0.5% SDS. Samples were resuspended in PAGE-buffer containing 143 mM β-mercaptoethanol, and the proteins were separated on 15% polyacrylamide gels and visualized with QuickBlue Protein stain (Lubio Science).

**Sample preparation for LC–MS analysis**. LtaA mutants were purified as described above, and concentrated to a concentration of 0.6 mg/ml. Purified LtaA was incubated for 1 h at RT in the dark in the absence or presence of 2 mM o-PDM. Afterward, 10 mM β-mercatoethanol was added to quench the cross-linker. In

total, 1–2 µg of either cross-linked or non-cross-linked LtaA protein were dissolved in 20 µl digestion buffer (0.02% of LMNG; 1 M urea; 0.1 M ammoniumbicarbonate; 10 mM tris(2-carboxyethyl) phosphine (TCEP); 15 mM chloroacetamide, pH = 8.5), reduced and alkylated for 1 h at 37 °C. Proteins were digested by incubation with either sequencing-grade modified trypsin (1/50, w/w; Promega, Madison, Wisconsin), chymotrypsin sequencing grade (1/50, w/w, Sigma-Aldrich) or lys-C (1/100, w/w, Wako) overnight at 37 °C. Then, the peptides were cleaned using iST cartridges (PreOmics, Munich) according to the manufacturer's instructions. Samples were dried under vacuum and dissolved in 0.1% formic acid solution at 0.5 pmol/µl. All samples were prepared in triplicates.

**Label-free targeted PRM LC–MS analysis of cysteine-containing peptides**. In a first step, parallel reaction-monitoring (PRM) assays[59] were generated for all the peptides of LtaA-WT and the peptides of the five different LtaA cysteine mutants, for each protease. These peptides include the reference peptide for normalization, which is shared for all mutants. Therefore, the specific peptide sequences were loaded into Skyline version 20.2.0.343 (https://brendanx-uw1.gs.washington.edu/labkey/project/home/software/Skyline/begin.view) and transitions were predicted using the integrated PROSIT algorithm for double- and triple-charged precursors. Then, protease and isoform-specific isolation mass lists were exported and used to generate specific targeted LC–MS analyses. This analysis was carried as described previously[60]. Chromatographic separation of peptides was carried out using an EASY nano-LC 1000 system (Thermo Fisher Scientific), equipped with a heated RP-HPLC column (75 µm × 30 cm) packed in-house with 1.9-µm C18 resin (Reprosil-AQ Pur, Dr. Maisch). Aliquots of 1 pmol total peptides were analyzed per LC–MS/MS run using a linear gradient ranging from 95% solvent A (0.15% formic acid, 2% acetonitrile) and 5% solvent B (98% acetonitrile, 2% water, 0.15% formic acid) to 30% solvent B over 90 minutes at a flow rate of 200 nl/min. Mass spectrometry analysis was performed on a Q-Exactive plus mass spectrometer equipped with a nanoelectrospray ion source (both Thermo Fisher Scientific) using a hybrid DDA (top5)/PRM LC–MS analysis. In detail, each MS1 scan was followed by high-collision-dissociation (HCD) of the precursor masses of the imported isolation list and the five most abundant precursor ions with dynamic exclusion for 20 s. For each mutant and protease, a specific LC–MS method was generated. Total cycle time was ~1 s. For MS1, 3e6 ions were accumulated in the Orbitrap cell over a maximum time of 100 ms and scanned at a resolution of 70,000 FWHM (at 200 $m/z$). Targeted MS2 scans were acquired at a target setting of 3e6 ions, accumulation time of 100 ms and a resolution of 35,000 FWHM (at 200 $m/z$) and a mass isolation window to 0.4 Th. MS1 triggered MS2 scans were acquired at a target setting of 1e5 ions, a resolution of 17,500 FWHM (at 200 $m/z$) and a mass isolation window of 1.4 Th. Singly charged ions and ions with unassigned charge state were excluded from triggering MS2 events. The normalized collision energy was set to 27% and one microscan was acquired for each spectrum. The acquired raw files were converted to mgf-file format using MSConvert (v 3.0, proteowizard) and searched using MASCOT (Matrix Science, Version: 2.4.1) against a decoy database containing normal and reverse sequences of the predicted SwissProt entries of *Staphylococcus aureus* (strain NCTC8325/PS 47, www.ebi.ac.uk, release date 2020/08/21). The five LtaA mutants and commonly observed contaminants (in total 6574 sequences) were generated using the SequenceReverser tool from the MaxQuant software (Version 1.0.13.13). The search criteria were set as follows: full tryptic specificity was required (cleavage after lysine or arginine residues); three missed cleavages were allowed; carbamidomethylation (C) was set as fixed modification and oxidation (M) as variable modification. The mass tolerance was set to 10 ppm for precursor ions and 0.02 Da for fragment ions. Then, Scaffold (version Scaffold_4.11.1, Proteome Software Inc., Portland, OR) was used to validate MS/MS-based peptide and protein identifications. Peptide identifications were accepted if they could be established at a probability greater than 97.0% by the Scaffold Local FDR algorithm. Protein identifications were accepted if they could be established at a probability higher than 99.0% to achieve an FDR less than 1.0% and contained at least two identified peptides. Protein probabilities were assigned by the Protein Prophet algorithm[61]. Proteins that contained similar peptides and could not be differentiated based on MS/MS analysis alone were grouped to satisfy the principles of parsimony. Subsequently, all raw files were imported into Skyline for protein/peptide quantification. To control for variation in sample amounts, all intensities were normalized against the four cysteine-free reference peptides. Only peptides that could be confidently identified by database searching were considered for quantification by PRM using the predicted transitions. Statistical analysis and ratio calculations to compare the relative abundance of the peptides between non-cross-linked and cross-linked peptides were performed in Excel. Histograms and *P* values were generated using Prism 9.

**Nanodiscs reconstitution**. Purified LtaA was reconstituted in MSP1D1 nanodiscs using a ratio of 2:5:115 (LtaA:MSP1D1:lipids) in a buffer containing 50 mM Tris-HCl, pH 8.0; 50 mM NaCl, and 10% glycerol (v/v). The lipid mixture used consists of 16:0-18:1 POPG:DAG (Avanti) in a 3:1 (w:w) ratio. Detergent was removed by adding Bio-beads SM2 (BioRad). After Bio-beads removal the mixture was centrifuged and loaded on a Superdex 200 Increase 10/300 GL (GE Healthcare) column equilibrated with buffer 50 mM Tris-HCl, pH 8.0; 50 mM NaCl. The peak corresponding to LtaA in nanodiscs was collected and used for mass photometry studies.

**Mass photometry assay**. Mass photometry experiments were performed using a Refeyn OneMP instrument operated at around 25 °C. Each measurement was made by mixing 2 µl of sample into 18 µl of buffer in a droplet contained by a Grace Bio-Labs CultureWell gasket (Sigma-Aldrich) on 24 × 50 mm area 170 ± 5 µm thickness coverglass (Marienfeld). Calibration of contrast to molecular mass was performed using Nativemark unstained protein standards (Thermo Fisher) at a final dilution of 1 in 500 in 0.2-µm filtered PBS. Samples of LtaA in nanodiscs, and of empty nanodiscs, were diluted to 100 nM final concentration in 50 mM Tris-HCl pH 8.0; 150 mM NaCl and contrast events were recorded for 120 s. The resulting movies were analyzed to obtain contrast event histograms in the software DiscoverMP 2.3.0 using five frames binning with motion correction, a Threshold-1 value of 2, and a Threshold-2 value of 0.25. After applying the molecular mass calibration, contrast event histograms were constructed using a bin-width of 8 kDa. Histograms from two consecutive 120 s measurements on the same sample were merged to include data with a lower protein deposition rate. To obtain molecular mass averages, histogram peaks were fitted to Gaussians using the manual peak-fitting procedure in the software.

**Docking of glycolipid**. Both 1,2-dihexadecanoic-3-O-(β-D-glucopyranosyl-1 → 6-O-β-D-glucopyranosyl-sn-glycerol) molecule and the headgroup (β-D-glucopyranosyl-1 → 6-O-β-D-glucopyranosyl-sn-Glycerol) were docked to the LtaA "repeat-swap" inward-facing models with Glide[62]. The initial coordinates of both full-length glycolipid and the headgroup were generated from 2D geometry in LigPrep[63]. The stereochemistry was corrected. Docking was performed over a search space of 45 × 45 × 45 Å³ covering the central cavity.

**S. aureus phenotypic complementation assay**. Generation of pLOW-ltaA and of *Staphylococcus aureus* NCTC8325 Δ*ltaA* genotype was previously described[27]. pLOW vector was used for construction of *ltaA* complentary strains. Point mutations were generated by extension overlap PCR, and then with restriction-ligation cloning using SalI and NotI cloned into pLOW vector[64]. For cloning purposes *E. coli* IM08B was used[65]. The sequence of the resulting constructs was confirmed by DNA sequencing (Microsynth). After conformation of the correct constructs, pLOW vector carrying ltaA-WT or point mutations were introduced into *S. aureus* NCTC8325 Δ*ltaA* by electrophoresis with erythromycin selection (5 µg/ml). *S. aureus* cells were grown in 3 ml of Luria-Bertani (LB) medium at 37 °C with 200 rpm until OD$_{600}$ of 0.3. For complementary strains containing a pLOW vector, a final concentration of 5 µg/ml erythromycin was added to the medium. For the serial dilutions, 5 µl of the original and its dilutions were spotted on LB agar plates buffered with sodium phosphate at pH 6.4 complemented with 0.1 mM IPTG. The plates were incubated overnight at 37 °C. Pictures were taken the next morning.

**Preparation of S. aureus membranes for LC–MS analysis**. *S. aureus* cells were grown in 3 ml LB medium at 37 °C with 200 rpm until OD$_{600}$ of 0.4. For complementary strains containing a pLOW vector, a final concentration of 5 µg/ml and 0.1 mM IPTG were added to the medium. After harvesting the cells were resuspended in 10 mM Tris pH 8.0; 1 mM EDTA; 25 µg/ml lysostaphin, and incubated for 0.5 h at 37 °C. Cells were further subjected to sonication, followed by the collection of membranes by ultracentrifugation. The membranes were resuspended in 100 mM Tris-HCl; 5% SDS; 10 mM tris(2-carboxyethyl) phosphine (TCEP). Samples were sonicated for 10 minutes, followed by shaking for 1 h at 37 °C with 500 rpm. To reduce and alkylate the disulfides a final concentration of 15 mM iodoacetamide was added, and the samples were incubated for 0.5 h in the dark at room temperature. Samples were loaded on S-trap Micro Spin column (Protifi). After washing, on-column peptide digestion was performed by addition of trypsin in 50 mM triethylammonium bicarbonate (TEAB) buffer, and incubation of 1 h at 47 °C. Digested peptides were collected by passing 50 mM triethylammonium bicarbonate (TEAB) buffer, 0.2% formic acid (w/v) in distilled water, and 0.2% formic acid (w/v) in 50% acetonitrile (v/v) through the column and dried in a SpeedVac (Labconco). Dried peptides were resuspended in 0.1% formic acid (w/v) and stored at −20 °C.

**Targeted PRM LC–MS analysis of LtaA-WT and mutants**. As a first step, PRM assays[59] for all possible peptides of LtaA with a length of 6 to 25 amino acids comprising double- and triple-charged precursor ions were created. Five peptides were identified to match the length and charge criteria, leading to ten PRM assays in total. These were used to identify LtaA membrane fractions of wild-type *S. aureus*. The setup of the µRPLC-MC system was previously described[60]. Mass spectrometry analysis was conducted using a Q-Exactive mass spectrometer with a nanoelectrospray ion source (both Thermo Fisher Scientific). Each MS1 scan was followed by high-collision-dissociation (HCD) of the ten LtaA precursor ions in PRM mode using a global isolation mass list. By applying strict identification criteria, three peptide ions of LtaA LTNYNTRPVK (2+ and 3+ ion) and MQDSSLNNYANHK (2+) were identified, and these were used for label-free PRM quantification. To control for protein variation between different samples, the total ion chromatography (only comprising peptide with two or more charges) was determined for each sample by label-free quantification using Progenesis QI (version 2.0, Waters) and then used to normalize the samples. The integrated peak

areas of the three peptide ions that were quantified by PRM were summed up and used for LtaA quantification.

**Modeling of inward-facing conformation.** The inward-facing conformation was modeled under the assumption of inverted repeats[48]. Sequence alignments between the two repeats of each domain of LtaA were performed. We structurally aligned R1D1 (residues 16–105) with R2D1 (residues 109–189), and R1D2 (residues 220–302) with R2D2 (residues 309–393) using the structure alignment program TMalign resulting in two pairs of alignments. These two pairs of alignments were then used together to build up the final pair-wise alignment between the LtaA sequence and a template in which the LtaA sequence repeats were rearranged in the order R1D2-R1D1-R2D2-R2D1. The initial sequence alignment was then refined by removing gaps in the transmembrane regions and in the secondary structure elements. In this step, one gap in the TM6 (between Phe220 and Pro221) and two gaps in TM8 (between Met294 and Ile295 and also between Leu296 and Ile297) in the alignment were removed. Also, residues Asp336 and Glu337 were moved left to remove a gap at the beginning of TM11. Further refinements were made to match the secondary structure as observed in the outward-open crystal structure. In particular, we aimed to maintain the helical regions in the template where possible, subject to the pseudo-symmetry between the two MFS transporter domains. In this step, a gap the loop region between TM11 and TM12 (between Thr368 and Asn369) was introduced to account for the correct orientation of TM12 helices.

We used the final refined alignment and the X-ray crystal structure of LtaA (PDB entry 6S7V (https://doi.org/10.1038/s41594-020-0425-5))[27] to construct the inward-facing model templates using Modeller 9v24. A template for modeling was constructed from the X-ray crystal structure of LtaA (PDB entry 6S7V) in which the coordinates of residues from R1D2, R1D1, R2D2, and R2D1 were placed as the first, second, third, and fourth segments of the template. Then, 100 initial models were generated. Next, we selected seven models with the highest MODELLER score and the best MolProbity[66] profile for further analysis. Then, we repacked the side chains using SCWRL4.0[67] and as a last step the models were energetically minimized after placing them in the lipid bilayer using the Gromacs steepest descent algorithm for 5000 steps[68]. To further validate the quality of the models, we assessed the stereochemistry. Evaluation of the model using MolProbity showed that the final minimized models have reasonable qualities (MolProbity score: 2.00-2.3, Ramachandran favored: 92.1–93.6%, and Ramachandran outliers: 0.8–1.90%) (Supplementary Table 1).

**Molecular dynamics simulations of inward-facing conformation models.** To study their dynamics, each of the seven optimized inward-facing models was placed in a heterogenous lipid bilayer (POPG (65%), diacylglycerol (20%), cardiolipin (10%), and gentiobiosyl-diacylglycerol (5%)) and then solvated in TIP3P water with 150 mM NaCl. The all-atom CHARMM36m force field was used for lipids, ions, and protein[69–71]. All simulations were performed using GROMACS 2019.6[68]. The starting systems were energy-minimized for 5000 steepest descent steps and equilibrated first for 1 ns of MD simulations in a canonical (NVT) ensemble and then for 7.5 ns in an isothermal-isobaric (NPT) ensemble under periodic boundary conditions. The initial restrains on the positions of non-hydrogen protein atoms were 4000 kJ·mol$^{-1}$·nm$^2$. During equilibration, these restraints were gradually released. Particle-mesh Ewald summation with cubic interpolation and a 0.12-nm grid spacing was used to treat long-range electrostatic interactions[72]. The time step was initially 1 fs and was then increased to 2 fs during the NPT equilibration. The LINCS algorithm was used to fix all bond lengths[73]. The constant temperature was set with a Berendsen thermostat[74], combined with a coupling constant of 1.0 ps. A semi-isotropic Berendsen barostat[74] was used to maintain a pressure of 1 bar. During production runs, the Berendsen thermostat and barostat were replaced by a Nosé–Hoover thermostat[75] and a Parrinello–Rahman barostat[76]. The unconstrained production trajectories were analyzed with Visual Molecular Dynamics (VMD)[77] and MDAnalysis package[78,79]. MD simulations of 500 ns were performed for each of the seven "repeat-swap" inward-facing models.

**Molecular dynamics simulations of outward-facing conformation.** The outward-facing structure of LtaA (PDB ID 6S7V [https://doi.org/10.1038/s41594-020-0425-5]) was embedded in a lipid bilayer composed of POPG-DAG-CL-gentiobiosyl-diacylglycerol using CHARMM-GUI[80]. The system was then solvated in TIP3P water with 150 mM NaCl. The all-atom CHARMM36m force field was used for lipids, ions, and protein[69–71]. All simulations were performed using GROMACS 2019.6[68]. Simulations were performed with similar protocols as described above for inward-facing models. The simulation of the outward-facing structure was performed for 1.2 µs.

**Molecular dynamics simulations of Alphafold inward-facing model.** The inward-facing model of LtaA generated by Alphafold[49] was embedded in a lipid bilayer composed of POPG-DAG-CL-gentiobiosyl-diacylglycerol using CHARMM-GUI[80]. The system was then solvated in TIP3P water with 150 mM NaCl. The all-atom CHARMM36m force field was used for lipids, ions, and protein[69–71]. All simulations were performed using GROMACS 2020.2[68]. The simulation was performed with similar protocols as described above for the inward-facing models. The simulation of the Alphafold model was performed for 1.1 µs.

**Reporting summary.** Further information on research design is available in the Nature Research Reporting Summary linked to this article.

## Data availability

The data that support the findings of this study are available from the corresponding author upon reasonable request. The mass spectrometry proteomics data have been deposited to the ProteomeXchange Consortium via the PRIDE[81] partner repository with the dataset identifier PXD031166. Source data are provided with this paper.

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

## Acknowledgements
We thank Prof. Jan-Willem Veening for providing us the *S. aureus* NCTC8325, *S. aureus* NCTC8325 Δ*ltaA*, *E. coli* IMO8B cells, and the pLOW vector. We thank Xiaochun Li Blatter for assistance in cell expression and membranes preparations. This work was supported by the Swiss National Science Foundation (SNSF) (PP00P3_198903 to C.P.), the Helmut Horten Stiftung (HHS) (to C.P.), and by the Max Planck Society and the German Research Foundation (SFB 807: Membrane Transport and Communication, to A.R.M. and G.H.). E.L. was funded by the Biozentrum International PhD Program and the HHS.

## Author contributions
E.L. performed in vitro and in vivo biochemical characterization of LtaA and variants. A.R.M. performed computational analysis. C.P. supervised the biochemical analysis. G.H. supervised computational analysis. A.S and E.L. performed mass spectrometry analysis. E.L., A.R.M., and C.P. analyzed the computational, structural, and functional data. C.P. conceived and directed the project. All authors contributed to manuscript writing and revision.

## Competing interests
The authors declare no competing interests.
