## [Peer Review File · Nature Communications]

Evidence for a trap-and-flip mechanism in a proton-dependent lipid transporterReviewers' Comments:

Reviewer #1:

Remarks to the Author:

The paper by Lambert and co-workers focuses on the transport mechanism of the glycolipid transporter LtaA, a proton-dependent MFS glycolipid transporter essential for lipoteichoic acids synthesis in the pathogen *Staphylococcus aureus*. Using a combination of modelling, conformational trapping, transport assays and MD simulations they propose a mechanism of how the glycolipid is transported. Their main conclusion is that glycolipids are transported using a canonical rocker-switch alternating access mechanism (termed here as a trap-and-flip mechanism) and that recognition for the lipid is dependent on the head-group. While the paper has some merits it is difficult to see the larger broader impact of this study.

Major concerns:

1. It is well established MFS transporters use a combination of local and global rearrangements to transport substrates across the membrane, which is typically referred to as a "rocker-switch" mechanism. That glycolipid substrates might not need to go through the same transport cycle, as every other MFS transporter, would be a controversial idea. I think this idea has come up from studies of ABC transporters, where it is unclear if lipid transporters need to undergo global rearrangements. However, transferring these ideas from ABC transporters to SLC transporters (like the MFS transporters) is something that you should be taken with extreme caution. Indeed, this study confirms that multiple conformational states are needed in the MFS transporter. While this is a good follow-up study, the results are, however, to be expected. Yet, I do not think the current version of the paper reflects the main-stream thinking in the MFS transporter field and implies using multiple states for a MFS lipid transporter is novel. Indeed, the authors conclude that the MFS transporters operate by a "trap-and-flip" mechanism, rather than just using the "rocker-switch" mechanism terminology that is typically used. I understand it is tempting to call the mechanism by some new terminology and some groups opt to use the "clamp-and-switch" rather than "rocker-switch" terminology to describe MFS transporter mechanisms. However, for consistency in the field, I think one should stick to the 3 types of alternating-access mechanism terminology (rocker-switch, rocking-bundle, elevator) rather than introduce new terms that add to confusion.

2. The inward-facing model has a high rmsd after being embedded into a lipid bilayer in MD simulations (around 6Å compared to 2-3Å for the experimental structure), which is worrying. I think the inward model is probably good enough to interpret the cross-linking data that confirms global bundle rearrangements are required for transport, but it lacks the robustness to start to describe detailed interactions.

Interestingly, I found that the alpha fold structure of LtaA is in the inward-facing conformation!

Judging from just how good these alpha fold structures have been, I would expect that the alpha fold structure of LtaA is going to be more reliable than the inward-facing model presented here (note that the residue numbering is 6 residues off from the numbering presented in the paper I couldn't work out where this discrepancy was coming from). With the availability of an excellent inward-facing LtaA structure, I would reanalyse the cross-linking data and re-run the MD simulations, which at the moment are more confirmatory rather than generating new insights.

In my opinion, for this paper to have a broader greater scientific impact, it needs to demonstrate the structural basis for glycolipid recognition and proton coupling. I think with the new structural information it should be possible to model how the sugar head group interacts with LtaA and what residues are important for recognition and proton-coupling. For example, in the alpha fold structure Arg35 is salt-bridged to Asp69 rather than the His64 residue. However, in the model here, Arg35 and Asp69 seem quite far apart. I think this interaction network could be important for de-tangling the

proton-coupled pathway. Also, most of the aromatic residues that I think are most likely to coordinate the sugar are not located at the bottom of the cavity. This implies that the tails would enter first rather than the headgroup, which also be different from the mechanism presented here. Clearly, its important to establish the residues required for substrate coordination and proton coupling.

3. The fluorescence-based transport assay is suboptimal with only 10% ACMA de-quenching, i.e., a relative drop fluorescence from 100 to 90% is very small. While this low activity might be good enough to conclude the protein is active calculate , I think it is not robust enough to compare mutants. Furthermore, to more robustly compare mutants, a relative K_m to WT should be compared for each mutant, rather than the fluorescence change at a single substrate concentration. I think to develop deeper mechanistic ideas other methods, such as SSM-based electrophysiology, might be required to supplement results this assay.

Reviewer #2:

Remarks to the Author:

Evidence for a 'trap-and-flip' mechanism in a proton-dependent lipid transporter

In this manuscript, Lambert et al., present an elegant model of the inward facing structure of LtaA and use cysteine crosslinking along with molecular dynamic simulations and transport assays (in vitro and in vivo) to investigate the mechanism of transport utilized by LtaA to flip glycolipids. Interestingly, the authors demonstrate that lipids enter via the TM2/11 lateral opening on the intracellular side but can be released via both the TM2/11 and TM5/8 lateral openings on the extracellular sides, and that central hydrophobic and hydrophilic pockets are important for LtaA mediated activity.

This well-written paper reveals several biochemically intriguing features of LtaA and MFS lipid transporters overall. However, in my opinion – intended as constructive criticism – some experiments would perhaps benefit by including more controls and analyses to form an unambiguous interpretation of precisely what is going on.

Major points/questions

In Fig. 3 and Sup Fig. 4 a 'dimer' band is sometimes present – is this protein known to function as a dimer? If dimer species are present, how do the authors confirm that the crosslinks are not between the individual protomers? In my opinion, the same experiment performed on single cysteine transfected and co-transfected mutants would be an important control to include that would answer this question.

Is a band also present for dimers with mPEG bound? It is hard to tell from Sup Fig. 4C, could staining be improved? Typically, with this sort of assay, bands are observed for both +1 mPEG5K and +2 mPEG5K (PMID 33597752 and PMID 24747941), however this is not seen here, or perhaps it is, could it be that the 'dimer' species is actually a +2 mPEG5K species? Evidence pointing towards this is highlighted in Sup for 4D where this band seemingly disappears in the presence of CuCl₂ and is absent for cysless – this may point to the actual case being that this "dimer species" is actually a +2 mPEG5K species and that CuCl₂ is a more effective crosslinking reagent than o-PDM.

Regarding K166C/I250E, do the authors believe it is natively crosslinked? I suggest performing the same experiment in the presence of the reducing agent DTT as I consider this a necessary control.

In summary I question, to a certain extent, the choice of o-PDM, and I emphasize the need for both a DTT control and single cysteine mutant controls (individually and co-transfected) for this assay.

For the fluorescence assays, WT was used as a control. In my opinion, cysless LtaA should be used as the control here rather than WT given the fact that the double cys mutants analyzed have been incorporated into the cysless construct (as opposed to the WT).

It would be informative if the authors could hypothesize as to why they didn't see lipid entry in the inward facing model, given LtaA is a lipid exporter. This seems to me an unexpected result and warrants further reflection, does it perhaps have anything to do with the [H⁺] in these simulations?

Can the authors suggest an explanation as to why V234W/I316W has low flipping activity in vivo but does not affect growth of *S. aureus*?

Throughout the manuscript, several references are made to a shared mechanism between LtaA and MFSD2A, whilst this is likely, the structure of MFSD2A shows lysolipid bound in the inward facing structure extending between TM5 and TM8 and MD simulations in this paper demonstrate that TM5 and TM8 is the major access point for lipid to exit the transporter into the inner leaflet. This appears to contrast with the results seen here for LtaA, where lipids mostly move through the TM2-TM11 cleft on the intracellular side. I suggest that the authors discuss this interesting difference.

The authors state that the results shown here demonstrate that LtaA encloses the full glycolipid substrate (line 288). However, in the MD snapshot shown in Fig. 4C it appears that the lipid tail extends into the lipid bilayer, and it was previously stated that just one tail of the glycolipid reaches the hydrophobic tail (line 237). While the docking shows glycolipids can bind in the central hydrophobic/hydrophilic pockets, I suggest including further evidence that the tail does not extend into the lipid bilayer (if this is the author's claim). Could the authors perhaps clarify this by providing information on the occupancy time of the lipid tail in the bilayer vs being fully engulfed? A supplementary video of the MD simulations may also be helpful.

The schematic representation shown in Figure 5F is rather similar to that previously published by the same authors with their structure of LtaA – my point being that it does not highlight or give justice to the new findings of this study. For example, it does not provide a representation for the finding that lipids enter via the TM2/11 lateral opening on the intracellular side but can be released via both the TM2/11 and TM5/8 lateral openings on the extracellular sides. It would also be informative for the hydrophobic and hydrophilic pockets to be highlighted in this schematic.

Minor points

Why were crosslinked fragments not seen with LCMS? Do they not fly?

Fig. 2A – It would be helpful to label distances between cysteines. They are shown graphically in Sup Fig. 1, but this information is simple to incorporate and important. Additionally, it would perhaps be more informative/accurate to state the distances as a table perhaps rather than as dots on the plot in Sup Fig. 1A.

In some of the figures, further details would be helpful, e.g., in the lower panel of Fig. 2C include o-PDM so that the reader doesn't have to go look at the top pane; also indicate +1 mPEG5K, +2 mPEG5K and no mPEG5K next to the arrows; in Fig. 3 say in the figure key that dashed line is - CuCl₂. Colour gradient keys of red-white-blue should be shown for electrostatic potentials for Fig. 5A and Sup Fig. 6B. These are of course minor points, but they would benefit the reader.

Yellow font is difficult to read – please consider changing the color.

Some gels overexposed – specifically on the right-hand side seem making them hard to interpret – e.g., lower panel of Fig. 2C and right panel of Sup Fig. 4C. It seems somewhat odd that the lower

panel of Fig. 2C and the center figure of Sup Fig. 4C are the same gel but with different exposure levels.

The membrane (in/out) should be indicated in Fig. 4A.

Not all residues mutated in Fig. 5 are shown in Fig. 5A, could they please be shown to assist the reading in interpreting the results?

I am curious as to why single mutants in the hydrophobic cavity were not tested. Presumably they have less of an effect on transport activity given the large size of the substrate – it would be interesting to see these results.

Reviewer #3:

Remarks to the Author:

This manuscript by Lambert et al reports on an experimental-computational study of the LtaA transporter, a proton-dependent lipid transporter of MFS superfamily. The MFS lipid transporters are generally understudied, which makes the current work significant. Also the combination of computational and experimental techniques in a synergistic manner as done in this work is of high interest.

This work is primarily based on a theoretical model of the inward-facing (IF) state of LtaA that is used as a guide to perform several experiments to validate the model and eventually hypothesize a transport mechanism for the LtaA. I think the approach is generally sound and the conclusions are generally reasonable; however, I have a number of questions that need to be answered or clarified before I can be sure of the soundness of the statements made.

(1) The initial model generated using the repeat-swap strategy from the available outward-facing (OF) structure is really what everything is built on in this work. However, the repeat-swap strategy is a general approach and as also vaguely discussed in the Methods section, many adjustments need to be made before having a reasonable model. The question is how good and reliable is this final model? One way of determining the stability of these models is of course MD, which is used here. However, very little information is provided to show the stability of the system. The only time series provided are RMSDs (Fig. S5B). It is not clear what the reference structure is for these RMSDs. I assume they are wrt the OF state. The fact that RMSD remains around 5 to 6 angstrom is not enough to establish stability. It is better to calculate the RMSD wrt to the initial models (i.e., at $t=0$) or even better at $t=150$ ns to see how much variation the systems have and whether the models are truly stable or not.

(2) On a related note, 150 ns is typically not enough to reach a stable conformation specially if one starts from a hypothetical model rather than a high-res crystal structure. If the RMSD analysis suggested above does not prove that the IF models are stable, I suggest to run longer simulations to arrive at more stable models.

(3) The authors state "we selected pairs of residues among the extracellular regions for which $C\beta-C\beta$ distances were less than 7 Å, but which present $C\beta-C\beta$ distances of over 12 Å in the outward-facing structure". It is not clear whether the three pairs identified are all the pairs that satisfy the above criteria or these three are only "selected" from a number of different options. This is of course quite important to know since if these are the only three pairs satisfying the stated criteria, the results provide a quite strong evidence for the validity of the IF models. Otherwise, the evidence would be considered much weaker and one would be interested to know about the other identified pairs as well.

Reviewer #1:

The paper by Lambert and co-workers focuses on the transport mechanism of the glycolipid transporter LtaA, a proton-dependent MFS glycolipid transporter essential for lipoteichoic acids synthesis in the pathogen *Staphylococcus aureus*. Using a combination of modelling, conformational trapping, transport assays and MD simulations they propose a mechanism of how the glycolipid is transported. Their main conclusion is that glycolipids are transported using a canonical rocker-switch alternating access mechanism (termed here as a trap-and-flip mechanism) and that recognition for the lipid is dependent on the head-group. While the paper has some merits it is difficult to see the larger broader impact of this study.

We thank the referee for her/his careful reading of our manuscript, and the constructive comments. According to her/his advice, we revised the manuscript and emphasized points of discussion that we expect to broaden the impact of our study.

Major concerns:

1. It is well established MFS transporters use a combination of local and global rearrangements to transport substrates across the membrane, which is typically referred to as a “rocker-switch” mechanism. That glycolipid substrates might not need to go through the same transport cycle, as every other MFS transporter, would be a controversial idea. I think this idea has come up from studies of ABC transporters, where it is unclear if lipid transporters need to undergo global rearrangements. However, transferring these ideas from ABC transporters to SLC transporters (like the MFS transporters) is something that you should be taken with extreme caution. Indeed, this study confirms that multiple conformational states are needed in the MFS transporter. While this is a good follow-up study, the results are, however, to be expected. Yet, I do not think the current version of the paper reflects the main-stream thinking in the MFS transporter field and implies using multiple states for a MFS lipid transporter is novel. Indeed, the authors conclude that the MFS transporters operate by a “trap-and-flip” mechanism, rather than just using the “rocker-switch” mechanism terminology that is typically used. I understand it is tempting to call the mechanism by some new terminology and some groups opt to use the “clamp-and-switch” rather than “rocker-switch” terminology to describe MFS transporter mechanisms. However, for consistency in the field, I think one should stick to the 3 types of alternating-access mechanism terminology (rocker-switch, rocking-bundle, elevator) rather than introduce new terms that add to confusion.

We agree that we have not discussed the transport mechanism of LtaA in the context of the ‘rocker-switch’ model for MFS transporters. Our rationale arose from studies of lipid transporters in which the terminologies ‘trap-and-flip’ or ‘credit-card’ are more commonly used when describing lipid transport processes. Discussing the mechanism using these terms is useful as they indicate whether the lipid is fully enclosed by the transporter (‘trap-and-flip’) or not (‘credit-card’). However, we agree with the referee that including a discussion on the ‘rocker-switch’ model is important, especially for researchers familiar with MFS transporters. Thus, we now mention the relation between the ‘rocker-switch’ model and the ‘trap-and-flip’ model (**lines 365-370**) and included new references (**Ref. 54 and 55**).

2. The inward-facing model has a high rmsd after being embedded into a lipid bilayer in MD simulations (around 6Å compared to 2-3Å for the experimental structure), which is worrying. I think the inward model is probably good enough to interpret the cross-linking data that confirms global bundle rearrangements are required for transport, but it lacks the robustness to start to describe detailed interactions.

We apologize for the confusion. The reported RMSD for the simulations of the inward-facing models was calculated relative to the outward-facing structure, which is expectedly higher than the RMSD for the simulation of the outward-facing structure itself. Thus, we now include the RMSD plots of the inward-facing models simulations relative to their starting model (**new Suppl. Fig. 5A**). The reported RMSDs are between 2-4 Å for all inward-facing models. In addition, we have also included RMSF plots (**new Suppl. Fig. 5D**) and a comparison of RMSD of all inward-facing models relative to the outward-facing structure (**new Suppl. Fig. 5E**).

Interestingly, I found that the alpha-fold structure of LtaA is in the inward-facing conformation! Judging from just how good these alpha-fold structures have been, I would expect that the alpha-fold structure of LtaA is going to be more reliable than the inward-facing model presented here. With the availability of an excellent inward-facing LtaA structure, I would re-analyse the cross-linking data and re-run the MD simulations, which at the moment are more confirmatory rather than generating new insights.

We agree that it is fascinating how AlphaFold predictions are revolutionizing the way structural biology advances, and we thank the reviewer for pointing out the inward-facing AlphaFold model, which we now included in our manuscript incrementing the insight of our study. According to the advice of the referee, we have performed a re-analysis of the cross-linking data, and also run MD simulations with the AlphaFold model. The new results are summarized below and changes to the manuscript are indicated.

(i) Re-analysis of cross-linking data: In the manuscript, we performed disulfide trapping of LtaA in inward-facing conformation by crosslinking of the following pairs of cysteine residues: F45C-T253C, A53C-T366C, and K166C-I250C. Crosslinking of these residues was confirmed by LC-MS and in-gel analysis (See **Fig. 2 and 4**), indicating that these residues are indeed in close vicinity when LtaA cycles through inward-facing conformations.

Analysis of the AlphaFold model, revealed similar C β -C β distances for two of these pairs (See table below). The deviation observed for the F45C-T253C pair is due to the AlphaFold model being in a more wide-open conformation. Since the AlphaFold model predicts a wide open inward-facing conformation of LtaA, relevant for glycolipid binding (see section starting at **line 194**), we have decided to include the discussion of this model in our manuscript (See **new Fig. 1D, 3A, 3B and 3D, and new Suppl. Fig. 5**). Changes to the results and discussion sections are indicated in **gray shading** in the main manuscript.

Cysteine pairs	approx. C β -C β distances (Å) 'Inward-facing models'	C β -C β distances (Å) 'AlphaFold model'
F45C-T253C	5.7	10.5
A53C-T366C	5.2	8.4
K166C-I250C	6.2	5.5

- (ii) MD simulations with AlphaFold model: In order to give further insight into the relevance of the AlphaFold model in the context of a membrane bilayer, we have performed MD simulations in a membrane composed of POPG (65%), diacylglycerol (20%), cardiolipin (10%), and gentiobiosyl-diacylglycerol (5%), resembling the membrane of *S. aureus*. Our results revealed that the cytoplasmic lateral openings of the AlphaFold model remain wide-open during the simulation (1.1 μ s), while the extracellular pathway remains closed. This indicates that the AlphaFold model represents an additional inward-facing state in which the central cavity of LtaA is more accessible from the cytoplasmic side. The results of this new MD simulation are now included in the manuscript (section starting at **line 194** and **new Fig. 1D, 3A, 3B and 3D, and new Suppl. Fig. 5**).

An additional significant result from this MD data is related to access of glycolipid substrate to the inward-facing cavity. The simulations reveal a molecule of glycolipid entering through the opening lined by TM5-TM8 (See section starting at **line 194**). An analysis on the implications of this finding is now included in the discussion section (**lines 370-388**).

We thank the reviewer again for pointing out the AlphaFold prediction. We believe that the analysis of this model has broadened the impact of our study!

Note that the residues numbering is 6 residues off from the numbering presented in the paper I couldn't work out where this discrepancy was coming from.

We would like to clarify that the discrepancy in residues numbering arises from a previous sequence of the annotated LtaA protein in uniprot (Q2FZP8), which contains six additional residues at the N-terminal domain. This disparity arises from some *S. aureus* strains containing these extra six additional residues in LtaA. For deposition of the LtaA structure (Zhang, et al. 2020), we adopted the sequence numbering with these additional six residues at the N-terminal domain. See also: [<https://www.uniprot.org/uniprot/Q2FZP8.txt?version=10>].

In my opinion, for this paper to broader greater scientific impact, it needs to demonstrate the structural basis for glycolipid recognition and proton coupling. I think with the new structural information it should be possible to model how the sugar head group interacts with LtaA and what residues are important for recognition and proton-coupling. For example, In the alpha-fold structure Arg35 is salt-bridged to Asp69 rather than the His64 residue. However, in the model here, Arg35 and Asp69 seem quite far apart. I think this interaction network could be important to for de-tangling the proton-coupled pathway. Also, most of the aromatic residues that I think are most likely to coordinate the sugar are not located at the bottom of the cavity. This implies that the tails would enter first rather than the headgroup, which also be different from the mechanism presented here. Clearly, its important to establish the residues required for substrate coordination and proton coupling.

We thank the reviewer for this comment. Indeed, establishing the residues responsible for substrate coordination and ion coupling are very important questions that help us understand the mechanism of a transporter in detail. We have already tackled these points in a previous publication (<https://www.nature.com/articles/s41594-020-0425-5>), where we elucidated a high-resolution structure of LtaA in an outward-facing state, showed docking of substrate into

the cavity, performed extensive mutagenesis and functional analysis of residues involved in substrate binding (headgroup recognition) and proton coupling. In that study, and in agreement with the hypothesis of the referee, we showed that indeed residues R35, D68, and E32 are involved in proton transport. Also, we showed that aromatic residues in the cavity are highly important for headgroup selectivity.

Regarding the Interaction of R35-D68, in the inward-facing models these two residues are indeed close together. The figure below shows that the distribution of the distance between the sidechain of these two residues is similar in the simulations of different inward-facing models, including the AlphaFold model. By contrast, the distribution for the simulation of the outward structure is narrower and the interaction is more stable. This observation further supports the hypothesis of the referee that this salt bridge might be important for proton coupling and transport.

3. The fluorescence-based transport assay is suboptimal with only 10% ACMA de-quenching, i.e., a relative drop fluorescence from 100 to 90% is very small. While this low activity might be good enough to conclude the protein is active calculate, I think it is not robust enough to compare mutants. Furthermore, to more robustly compare mutants, a relative K_m to WT should be compared for each mutant, rather than the fluorescence change at a single substrate concentration. I think to develop deeper mechanistic ideas other methods, such as SSM-based electrophysiology, might be required to supplement results this assay.

In the fluorescence-based assay, the proportion of ACMA quenched is related to the rate of the transport system under study. Lipid transporters are slow in contrast to transporters of water-soluble substrates. Thus, our ACMA quenching experiments are representative of this observation. Furthermore, even for very fast proteins (e.g. Cl⁻/H⁺ exchanger), ACMA quenching does not fall below 60%, please see the following studies:

<https://www.science.org/doi/10.1126/science.1195230>

<https://www.pnas.org/content/109/29/11699.long>

Thus, our ACMA experiments agree with what other labs have reported when using this methodology. In addition, we would like to point out that the signal-to-noise ratio of our

fluorescence recordings is very high, thus allowing a clear differentiation between non-active and active mutants as seen in **Fig. 4**.

Regarding the use of other methodologies, such as SSM-based electrophysiology, it is unfortunately not possible to apply such technique for the study of LtaA. This is due to the nature of the substrate (lipid embedded in a membrane), which cannot be dissolved in aqueous buffers for initiation of transport, as required for SSM-electrophysiology or many other transport assays.

Reviewer #2:

In this manuscript, Lambert et al., present an elegant model of the inward facing structure of LtaA and use cysteine crosslinking along with molecular dynamic simulations and transport assays (in vitro and in vivo) to investigate the mechanism of transport utilized by LtaA to flip glycolipids. Interestingly, the authors demonstrate that lipids enter via the TM2/11 lateral opening on the intracellular side but can be released via both the TM2/11 and TM5/8 lateral openings on the extracellular sides, and that central hydrophobic and hydrophilic pockets are important for LtaA mediated activity. This well-written paper reveals several biochemically intriguing features of LtaA and MFS lipid transporters overall. However, in my opinion – intended as constructive criticism – some experiments would perhaps benefit by including more controls and analyses to form an unambiguous interpretation of precisely what is going on.

We thank the referee for her/his careful reading of our manuscript, and the constructive comments. According to her/his advices, we revised the manuscript as much as we could, including new experiments that we hope are adequate to answer the concerns raised by the referee and that provide new insights into the mechanism of LtaA.

Major points/questions:

In Fig. 3 and Sup Fig. 4 a 'dimer' band is sometimes present – is this protein known to function as a dimer? If dimer species are present, how do the authors confirm that the crosslinks are not between the individual protomers? In my opinion, the same experiment performed on single cysteine transfected and co-transfected mutants would be an important control to include that would answer this question.

We thank the reviewer for raising this point. We agree with the reviewer that the dimer band might be interpreted as a potential artifact of interprotomer crosslinking. Such crosslink might indeed spontaneously occur by (i) absence of reducing agent and (ii) if free cysteines are well exposed to the solvent. In order to prevent interprotomer crosslinking, all the mutants were purified in the presence of 5 mM β -mercaptoethanol (see **Methods** section, **lines 397-416**). In addition, proteoliposomes reconstitution, as well as storage was done with buffer containing 2 mM β -mercaptoethanol (**lines 453-460**). Only shortly before performing crosslinking assays, the buffer was exchanged to buffer without β -mercaptoethanol (**lines 501-502**). Furthermore, all the gels (made after treatment with mPEG5k) were run with samples resuspended in PAGE-buffer containing reducing agent in order to avoid crosslink between unreactive free cysteines (**lines 510-511**). We have now included more details about this in the lines mentioned above.

In order to clarify this point further, we have followed the instructions of the reviewer and cloned, expressed, purified, reconstituted, and performed gel analysis of single, double, and tetra cysteine mutants after treatment of samples with 5 mM DTT. This treatment would ensure that all potential interprotomer disulfide bridges would be disrupted before SDS-PAGE (**see Figure below**). Our results show that independently of these samples being treated under harsh reducing conditions, the dimer specie is still clearly observed even for LtaA-WT and LtaA

cys-less. Thus, we conclude that the dimer band is not related to interprotomer crosslinking and rather shows other kind of interactions between LtaA proteins during in gel analysis. We now include a small discussion about this band in the manuscript (**lines 176-180**).

SDS-PAGE analysis of cysteine mutants after treatment with 5 mM DTT. Gels show samples from different experiments.

In agreement with the reviewer, we further investigated the question about whether LtaA is a dimer in a membrane. To assess this, we reconstituted LtaA in lipid nanodiscs and performed mass-photometry assays where we compared the masses of nanodiscs containing LtaA and nanodiscs without LtaA (see **Figure below -new Suppl. Fig. 2D**). Our results show that LtaA is a monomer in the membrane of nanodiscs (Mass photometry estimated LtaA mass= 46kDa, LtaA theoretical Mass = 44.2 kDa). However, whether LtaA is a monomer or dimer in the membrane of *S. aureus* cells remains to be explored in a future study as this is out of the scope of this manuscript.

Mass photometry of empty (A) and LtaA-reconstituted (B) nanodiscs. A) Empty nanodiscs: molecular mass 139 kDa. B) LtaA nanodiscs: molecular mass 185 kDa. The molecular mass was determined from the mean of the Gaussian distribution. Apparent molecular mass difference: 46 kDa.

Is a band also present for dimers with mPEG bound? It is hard to tell from Sup Fig. 4C, could staining be improved? Typically, with this sort of assay, bands are observed for both +1 mPEG5K and +2 mPEG5K (PMID 33597752 and PMID 24747941), however this is not seen here, or perhaps it is, could it be that the 'dimer' species is actually a +2 mPEG5K species? Evidence pointing towards this is highlighted in Sup for 4D where this band seemingly disappears in the presence of CuCl₂ and is absent for cysless – this may point to the actual case being that this “dimer species” is actually a +2 mPEG5K species and that CuCl₂ is a more effective crosslinking reagent than o-PDM.

Please see the answer to the first point. There, we show that the band at ~52kDa is present even in the absence of mPEG5K, demonstrating that this is not due to a +2 addition of mPEG5k. Also, the same band is present in the cys-less and WT proteins, for which it is not possible to react with 2 molecules of mPEG5k.

In order to clarify this point further, we performed PEGylation reactions of the double cysteine mutants at incremental time points (see Figure below).

SDS-PAGE analysis of cysteine mutants treated with mPEG5k at different reaction times. +CuCl₂ indicates pre-treatment with CuCl₂.

Our results show that there is no appearance of an additional PEGylation band besides the one showing between 37 and 52 kDa. This same band is also clearly the only band disappearing after treatment with reducing agent. Thus, our results suggest that under our experimental conditions PEGylation occurs at a very fast rate and that the band between 37 and 52 kDa correspond to +2mPEG5k species. This is not striking as in contrast to the cysteine pairs present in the proteins that the referee mentioned in references PMID 33597752 and PMID 24747941, the cysteine pairs introduced in the LtaA variants are much more exposed to the solvent (see Figure below). Thus, it is expected for our cysteine mutants to react very quickly with mPEG5k.

Regarding K166C/I250C, do the authors believe it is natively crosslinked? I suggest performing the same experiment in the presence of the reducing agent DTT as I consider this a necessary control.

We thank the reviewer for the comment. As mentioned above all variants were purified in the presence of reducing agent, which was removed shortly before performing crosslinking experiments. Thus, we did not expect the mutant K166C/I250C to show native crosslinking. However, following the advice of the reviewer, we have performed additional experiments where we pre-treated the mutant K166C/I250C, as well as the other double and tetra cysteine mutants with 5 mM DTT before crosslinking and subsequent PEGylation (see **Figure below - new Fig. 2C and new Suppl. Fig. 4C**). Our results still show a slightly weaker PEGylation band for the variant K166C/I250C, which clearly disappears after crosslinking with o-PDM. Thus, the weaker band is not due to native crosslinking but rather a mutant-specific staining effect.

SDS-PAGE analysis of cysteine mutants treated with mPEG5k after crosslinking with o-PDM (samples pre-treated with 5mM DTT)

In summary I question, to a certain extent, the choice of o-PDM, and I emphasize the need for both a DTT control and single cysteine mutant controls (individually and co-transfected) for this assay.

In agreement with the reviewer, we provided here the necessary controls as mentioned above.

For the fluorescence assays, WT was used as a control. In my opinion, cysless LtaA should be used as the control here rather than WT given the fact that the double cys mutants analyzed have been incorporated into the cysless construct (as opposed to the WT).

We thank the reviewer for pointing this out. We have now included the corresponding measurement with cys-less LtaA in the **new Fig. 4A and 4C**.

It would be informative if the authors could hypothesize as to why they didn't see lipid entry in the inward facing model, given LtaA is a lipid exporter. This seems to me an unexpected result and warrants further reflection, does it perhaps have anything to do with the [H⁺] in these simulations?

As pointed out by reviewer #1, while the manuscript was under review, a model of LtaA in a wide-open inward facing conformation was generated by AlphaFold. MD simulations of this

model in a lipid membrane show evidence of glycolipids entering the inward facing cavity of LtaA. These new results are now included in our manuscript.

Can the authors suggest an explanation as to why V234W/I316W has low flipping activity *in vivo* but does not affect growth of *S. aureus*?

The *in vitro* activity of this mutant is about two-fold lower than that of LtaA WT. We hypothesize that under *in vivo* conditions, the remaining 50% activity is enough to supply the lipoteichoic acid pathway without a dramatic impact on cell growth. This hypothesis is now included in **lines 295-296**.

Throughout the manuscript, several references are made to a shared mechanism between LtaA and MFSD2A, whilst this is likely, the structure of MFSD2A shows lysolipid bound in the inward facing structure extending between TM5 and TM8 and MD simulations in this paper demonstrate that TM5 and TM8 is the major access point for lipid to exit the transporter into the inner leaflet. This appears to contrast with the results seen here for LtaA, where lipids mostly move through the TM2-TM11 cleft on the intracellular side. I suggest that the authors discuss this interesting difference.

We thank the reviewer for this question. As mentioned above, reviewer #1 pointed towards a wide-open inward facing model generated by AlphaFold. MD simulations of this model in a lipid membrane show evidence of glycolipids entering the inward facing cavity of LtaA through the lateral opening lined by TM5-TM8. In agreement with what has been shown for MFSD2A. The results of MD analysis with this model and re-analysis of our experimental data is now included in our manuscript (see the two sections starting at **line 194 and 224**; and **new Fig. 1D, 3A, 3B and 3D, and new Suppl. Fig. 5**). An analysis on the implications of this finding is now included in the discussion section (**lines 370-388 and new Fig. 5F**).

These new results agree with the lateral opening lined by TM5-TM8 being relevant for lipid transport, similarly to what has been reported for MFSD2A and correctly pointed out by the referee. By contrast, we postulate that the opening lined by TM2-TM11 is important for ion-dependent gating as similarly suggested for MFSD2A (**Ref. 22**).

The authors state that the results shown here demonstrate that LtaA encloses the full glycolipid substrate (line 288). However, in the MD snapshot shown in Fig. 4C it appears that the lipid tail extends into the lipid bilayer, and it was previously stated that just one tail of the glycolipid reaches the hydrophobic tail (line 237). While the docking shows glycolipids can bind in the central hydrophobic/hydrophilic pockets, I suggest including further evidence that the tail does not extend into the lipid bilayer (if this is the author's claim). Could the authors perhaps clarify this by providing information on the occupancy time of the lipid tail in the bilayer vs being fully engulfed? A supplementary video of the MD simulations may also be helpful.

We thank the reviewer for raising this important issue. We also agree that a snapshot is not the best way to show a dynamic process. Now, we have added two movies to the supplementary materials. Movie 1 shows the lipids entering into the cavity in the outward-facing conformation. Movie 2 shows the entrance of a glycolipid into the cavity from the TM5-TM8 lateral opening in the inward-facing model generated by AlphaFold.

The schematic representation shown in Figure 5F is rather similar to that previously published by the same authors with their structure of LtaA – my point being that it does not highlight or give justice to the new findings of this study. For example, it does not provide a representation for the finding that lipids enter via the TM2/11 lateral opening on the intracellular side but can be released via both the TM2/11 and TM5/8 lateral openings on the extracellular sides. It would also be informative for the hydrophobic and hydrophilic pockets to be highlighted in this schematic.

In agreement with the reviewer and in light of the new results, we have now improved this mechanistic model (see new Fig. 5F).

Minor points:

Why were crosslinked fragments not seen with LCMS? Do they not fly?

We thank the reviewer for pointing out this relevant question. LC-MS analysis of cross-linked peptides has become routine over the last years and is also used in our facility at BZ. However, as an integral membrane protein, LtaA has very few cleavage sites for standard proteases used in LC-MS based proteomics (like trypsin or lys-C), leading to very large peptides. We used additional proteases (Glu-C and Chymotrypsin) to generate shorter peptides suited for LC-MS analysis to identify at least one of the two cysteine containing peptides. However, it was not possible to find conditions that would generate good peptides covering both cysteine residues using the same sample preparation workflow, which unfortunately led to crosslinked peptides that were too large for direct LC-MS analysis.

Fig. 2A – It would be helpful to label distances between cysteines. They are shown graphically in Sup Fig. 1, but this information is simple to incorporate and important. Additionally, it would perhaps be more informative/accurate to state the distances as a table perhaps rather than as dots on the plot in Sup Fig. 1A.

We have introduced these distances.

In some of the figures, further details would be helpful, e.g.,
in the lower panel of Fig. 2C include o-PDM so that the reader doesn't have to go look at the top pane;
also indicate +1 mPEG5K, +2 mPEG5K and no mPEG5K next to the arrows;
in Fig. 3 say in the figure key that dashed line is - CuCl₂.
Colour gradient keys of red-white-blue should be shown for electrostatic potentials for Fig. 5A and Sup Fig. 6B. These are of course minor points, but they would benefit the reader.

Thanks for these important points. These changes have been now included.

Yellow font is difficult to read – please consider changing the color.

This has been changed now.

Some gels overexposed – specifically on the right-hand side seem making them hard to interpret – e.g., lower panel of Fig. 2C and right panel of Sup Fig. 4C. It seems somewhat odd that the lower panel of Fig. 2C and the center figure of Sup Fig. 4C are the same gel but with different exposure levels.

We have rerun this gel, and obtained better staining. This is now included in a **new Fig. 2C and Suppl. Fig. 4C**.

The membrane (in/out) should be indicated in Fig. 4A.

This has been changed now (**new Fig. 3A**).

Not all residues mutated in Fig. 5 are shown in Fig. 5A, could they please be shown to assist the reading in interpreting the results?

This has been changed now.

I am curious as to why single mutants in the hydrophobic cavity were not tested. Presumably they have less of an effect on transport activity given the large size of the substrate – it would be interesting to see these results.

We thank the referee for this question. Due to the size of the cavity, we wanted to increase the chances to modify hydrophilicity significantly enough to see an effect in *in vivo* and *in vitro* assays.

Reviewer #3:

This manuscript by Lambert et al reports on an experimental-computational study of the LtaA transporter, a proton-dependent lipid transporter of MFS superfamily. The MFS lipid transporters are generally understudied, which makes the current work significant. Also the combination of computational and experimental techniques in a synergistic manner as done in this work is of high interest. This work is primarily based on a theoretical model of the inward-facing (IF) state of LtaA that is used as a guide to perform several experiments to validate the model and eventually hypothesize a transport mechanism for the LtaA. I think the approach is generally sound and the conclusions are generally reasonable; however, I have a number of questions that need to be answered or clarified before I can be sure of the soundness of the statements made.

We thank the referee for the carefully reading our manuscript and her/his kind comments on our work.

(1) The initial model generated using the repeat-swap strategy from the available outward-facing (OF) structure is really what everything is built on in this work. However, the repeat-swap strategy is a general approach and as also vaguely discussed in the Methods section, many adjustments need to be made before having a reasonable model. The question is how good and reliable is this final model? One way of determining the stability of these models is of course MD, which is used here. However, very little information is provided to show the stability of the system. The only time series provided are RMSDs (Fig. S5B). It is not clear what the reference structure is for these RMSDs. I assume they are wrt the OF state. The fact that RMSD remains around 5 to 6 angstrom is not enough to establish stability. It is better to calculate the RMSD wrt to the initial models (i.e., at t=0) or even better at t=150 ns to see how much variation the systems have and whether the models are truly stable or not.

We are sorry for the confusion regarding the RMSD plots. The RMSD data shown for the simulations are relative to the outward-facing structure. In the revised version of the manuscript, in agreement with the comments of the reviewer, we now provide further evidence for the stability of our models. We have calculated RMSDs relative to the initial models (see **new Suppl. Fig. 5**). We also show the root mean square fluctuations (RMSFs) of the models in the **new Suppl. Fig. 5**, calculated for each simulation run based on the average structure. From all three RMSD and RMSF plots, we can conclude that our inward-facing models are quite stable. We also extend our methods section about the 'repeat-swap' approach and discussed the steps taken in this process in more detail (**see lines 659-725**).

(2) On a related note, 150 ns is typically not enough to reach a stable conformation specially if one starts from a hypothetical model rather than a high-res crystal structure. If the RMSD analysis suggested above does not prove that the IF models are stable, I suggest to run longer simulations to arrive at more stable models.

We thank the reviewer for pointing this out. We have now extended the time of our MD simulations to 500 ns for each of the 7 inward models and still all the models are stable in the

membrane environment during the simulations. We have now included this information in our manuscript and included a **new suppl. Figure 5**.

(3) The authors state "we selected pairs of residues among the extracellular regions for which C β -C β distances were less than 7 Å, but which present C β -C β distances of over 12 Å in the outward-facing structure". It is not clear whether the three pairs identified are all the pairs that satisfy the above criteria or these three are only "selected" from a number of different options. This is of course quite important to know since if these are the only three pairs satisfying the stated criteria, the results provide a quite strong evidence for the validity of the IF models. Otherwise, the evidence would be considered much weaker and one would be interested to know about the other identified pairs as well.

We thank the reviewer for this comment. The pairs of residues described in the manuscript were selected not only based on the analysis of C β -C β distances, but also considering their accessibility to crosslinking agents (avoiding residues deeply buried) and avoiding those present in flexible loops, which might decrease the success of crosslinking. We apologize for not specifying this information in the previous version of the manuscript. This is now included in the new version in **lines 137-138**.

As indicated in the figure below, three additional pairs of residues in inward-facing LtaA fulfill the criteria of displaying C β -C β distances of less than 7 Å in this state but longer than 12 Å in the outward-facing state (1. A55-G363, 6.1 Å; 2. V49-S248, 6.5 Å; and 3. N162-I257, 4.1 Å). However, these pairs were not selected based on the considerations mentioned above. They are either buried and likely hardly accessible, or found in a flexible loop.

Extracellular view of inward-facing model of LtaA and additional potential residues for exchange to cysteine.

Reviewers' Comments:

Reviewer #1:

Remarks to the Author:

I thank the author for their responses and I think the paper is improved overall. However, I still have major technical concerns on the activity measurements using ACMA, e.g., Figure 4A. We use these pH sensitive dyes routinely in the lab and signals this low have to be taken with caution. The responses with valinomycin drops the signal 5% in the protein free liposomes. With protein, the signal drops to 10% dequenching. So the "real" signal is only a change in 5%. Also empty liposomes and proteoliposomes will have different backgrounds. I think more data is needed to prove the validity of these measurements since the signal is so low. Indeed, these measurements are only technical repeats. However, the absolute signal for WT can easily vary by more than 10% on independent protein reconstitutions, due to the stochastic nature of protein incorporation efficiency into liposomes.

These additional experiments should be straightforward to perform with your setup and would alleviate my concerns.

1. Show with WT protein that the signal increases with increasing potassium-diffusion potential (mV)
2. The protein should be oxidised during purification. The comparison between plus and minus CuCl₂ is not really the best control. Please demonstrate the signal difference for the cross-links mutants (or at least one of them) with and without DTT addition. Given the small signal I think this is the "cleanest" experiment to validate the activity measurements.

In addition to the above in the introduction it is written "Two different general models of transporter-catalyzed lipid translocation have been proposed in the past". I would make it clearer that these two different general lipid models have been proposed for non-MFS transporters. At the end of the day, the final mechanism (termed here a "trap-and-flip") is the rocker-switch alternating-access mechanism that all MFS transporters use. Please note that the additional references added for the rocker-switch mechanism considered at the time that the two bundles moved as rigid bodies. The "rocker-switch" has been redefined to encompass both local and global conformational changes and you could include a more recent review for this here, e.g. ref. 4 would be fine.

Reviewer #2:

Remarks to the Author:

I thank the Authors for fulfilling all of my requests and answering all of my questions.

I have no further comments and am satisfied with the changes made and explanations provided.

Reviewer #3:

Remarks to the Author:

The authors have addressed all my concerns.

Reviewer #1:

I thank the author for their responses and I think the paper is improved overall. However, I still have major technical concerns on the activity measurements using ACMA, e.g., Figure 4A. We use these pH sensitive dyes routinely in the lab and signals this low have to be taken with caution. The responses with valinomycin drops the signal 5% in the protein free liposomes. With protein, the signal drops to 10% dequenching. So the “real” signal is only a change in 5%. Also empty liposomes and proteoliposomes will have different backgrounds. I think more data is needed to prove the validity of these measurements since the signal is so low. Indeed, these measurements are only technical repeats. However, the absolute signal for WT can easily vary by more than 10% on independent protein reconstitutions, due to the stochastic nature of protein incorporation efficiency into liposomes.

These additional experiments should be straightforward to perform with your setup and would alleviate my concerns.

We thank the referee for her/his comments. According to her/his advice, we revised the manuscript, clarified concerns, and show additional experimental evidence that answer the reviewer’s concerns about the ACMA assay.

1. Show with WT protein that the signal increases with increasing potassium-diffusion potential (mV)

We thank the referee for the comment. According to her/his advice, we have performed proton transport assays in the presence of different potassium-diffusion potentials (see **Figure below**), generated by adjusting the KCl concentration inside and outside of LtaA-WT proteoliposomes. Proton transport was initiated by the addition of valinomycin as indicated in the manuscript. Our results show that, as expected, changes in the potassium-diffusion potential affect proton transport. The more negative the potential, the faster protons are imported, thus, further ACMA quenching is observed.

2. The protein should be oxidised during purification. The comparison between plus and minus CuCl₂ is not really the best control. Please demonstrate the signal difference for the cross-links mutants (or at least one of them) with and without DTT addition Given the small signal I think this is the “cleanest” experiment to validate the activity measurements.

We would like to kindly remind the referee about our discussion with reviewer #2 (previous round of revision). In summary, such crosslinking (oxidation) of cysteine pairs might indeed

spontaneously occur during purification and/or reconstitution if (i) a reducing agent is absent and (ii) if free cysteines are well exposed to the solvent. In order to prevent this, the cysteine mutants were purified in harsh reducing conditions (5 mM β -mercaptoethanol -see **Methods** section, **lines 396-415**-). In addition, proteoliposomes reconstitution, as well as storage was done with buffer containing 2 mM β -mercaptoethanol (**lines 452-459**). Only shortly before performing crosslinking assays, the buffer was exchanged to buffer without β -mercaptoethanol (**lines 500-501**). Thus, all the activity assays minus CuCl₂ indicate samples where cysteines are not crosslinked. Furthermore, this claim is supported by our LC-MS analysis (**Fig. 2A,B**) and in-gel analysis with mPEG5K (**Fig. 2C and 3D**).

In addition to the above in the introduction it is written “Two different general models of transporter-catalyzed lipid translocation have been proposed in the past”. I would make it clearer that these two different general lipid models have been proposed for non-MFS transporters. At the end of the day, the final mechanism (termed here a “trap-and-flip”) is the rocker-switch alternating-access mechanism that all MFS transporters use.

We added this clarification to the introduction part. **See line 77.**

Please note that the additional references added for the rocker-switch mechanism considered at the time that the two bundles moved as rigid bodies. The “rocker-switch” has been redefined to encompass both local and global conformational changes and you could include a more recent review for this here, e.g. ref. 4 would be fine.

We thank the reviewer for this comment. Reference 4 has been added to the discussion about the “rocker-switch” model. **See lines 367-369.**

Reviewers' Comments:

Reviewer #1:

Remarks to the Author:

I thank the authors for these additional experiments and additions to the methods section. I have no further concerns.